# Safety and feasibility of an *in situ* vaccination and immunomodulatory targeted radionuclide combination immuno-radiotherapy approach in a comparative (companion dog) setting

Kara Magee[1], Ian R. Marsh[2], Michelle M. Turek[3], Joseph Grudzinski[2], Eduardo Aluicio-Sarduy[2], Jonathan W. Engle[2], Ilene D. Kurzman[1], Cindy L. Zuleger[4,5], Elizabeth A. Oseid[6], Christine Jaskowiak[7], Mark R. Albertini[4,5,8], Karla Esbona[9], Bryan Bednarz[2,7], Paul M. Sondel[5,10,11], Jamey P. Weichert[5,7], Zachary S. Morris[5,9], Reinier Hernandez[2,5,7]*, David M. Vail[1,5]*

**1** Department of Medical Sciences, School of Veterinary Medicine, University of Wisconsin-Madison, Madison, Wisconsin, United States of America, **2** Department of Medical Physics, School of Medicine and Public Health, University of Wisconsin-Madison, Madison, Wisconsin, United States of America, **3** Department of Surgical Sciences, School of Veterinary Medicine, University of Wisconsin-Madison, Madison, Wisconsin, United States of America, **4** Department of Medicine, School of Medicine and Public Health, University of Wisconsin-Madison, Madison, Wisconsin, United States of America, **5** Carbone Cancer Center, University of Wisconsin-Madison, Madison, Wisconsin, United States of America, **6** Office of Environment, Health and Safety, University of Wisconsin-Madison, Madison, Wisconsin, United States of America, **7** Department of Radiology, School of Medicine and Public Health, Madison, Wisconsin, United States of America, **8** The Medical Service, William S. Middleton Memorial Veterans Hospital, Madison, Wisconsin, United States of America, **9** Department of Pathology and Laboratory Medicine, School of Medicine and Public Health, Madison, Wisconsin, United States of America, **10** Department of Human Oncology, School of Medicine and Public Health, Madison, Wisconsin, United States of America, **11** Department of Pediatrics, School of Medicine and Public Health, University of Wisconsin-Madison, Madison, Wisconsin, United States of America

* hernandez6@wisc.edu (RH); david.vail@wisc.edu. (DMV)

## Abstract

### Rationale

Murine syngeneic tumor models have revealed efficacious systemic antitumor responses following primary tumor *in situ* vaccination combined with targeted radionuclide therapy to secondary or metastatic tumors. Here we present studies on the safety and feasibility of this approach in a relevant translational companion dog model (n = 17 dogs) with advanced cancer.

### Methods

The three component of the combination immuno-radiotherapy approach were employed either separately or in combination in companion dogs with advanced stage cancer. *In situ* vaccination was achieved through the administration of hypofractionated external beam radiotherapy and intratumoral hu14.18-IL2 fusion immunocytokine injections to the index tumor. *In situ* vaccination was subsequently combined with targeted radionuclide therapy

**Data Availability Statement:** All relevant data are within the manuscript and its Supporting Information files.

**Funding:** Financial Support: Support provided by the U01CA233102 (NIH/HHS), the University of Wisconsin Carbone Cancer Center Support Grant (P30 CA014520, NIH/HHS), the Barbara A. Suran Comparative Research Endowment, the Merit Review Award I01 BX003916 from the Biomedical Laboratory Research and Development Service of the United States (U.S.) Department of Veterans Affairs, P50026787 (NIH/NIDCR). This material includes studies in client-owned companion dogs supported with resources and the use of facilities at the William S. Middleton Memorial Veterans Hospital, Madison, WI. The contents do not represent the views of the U.S. Department of Veterans Affairs or the United States Government.

**Competing interests:** I have read the journal's policy and the authors of this manuscript have the following compeXng interests: ZM – Scientific Advisory Board Member and equity options, Archeus Technologies, Scientific Advisory Board Member and equity options, Seneca Therapeutics, Research Material support (reagents) from Bristol-Myers Squibb, AstraZeneca, Nektar Therapeutics, Apeiron Biologics, XRAd Therapeutics. BB and JG has ownership interest in Voximetry Inc (Middleton, WI). MRA - Research collaborations through the University of Wisconsin with Bristol Myers Squibb (Redwood City, CA and Princeton, NJ) and with Apeiron Biologics (Vienna, Austria). JW is a co-founder and CSO of Archeus Technologies, Inc (Madison, WI). This does not alter our adherence to PLOS ONE policies on sharing data and materials.

using a theranostic pairing of IV $^{86}$Y-NM600 (for PET imaging and subject-specific dosimetry) and IV $^{90}$Y-NM600 (therapeutic radionuclide) prescribed to deliver an immunomodulatory 2 Gy dose to all metastatic sites in companion dogs with metastatic melanoma or osteosarcoma. In a subset of dogs, immunologic parameters preliminarily assessed.

## Results

The components of the immuno-radiotherapy combination were well tolerated either alone or in combination, resulting in only transient low grade (1 or 2) adverse events with no dose-limiting events observed. In subject-specific dosimetry analyses, we observed $^{86}$Y-NM600 tumor:bone marrow absorbed-dose differential uptakes $\geq 2$ in 4 of 5 dogs receiving the combination, which allowed subsequent safe delivery of at least 2 Gy $^{90}$Y-NM600 TRT to tumors. NanoString gene expression profiling and immunohistochemistry from pre- and post-treatment biopsy specimens provide evidence of tumor microenvironment immunomodulation by $^{90}$Y-NM600 TRT.

## Conclusions

The combination of external beam radiotherapy, intratumoral immunocytokine, and targeted radionuclide immuno-radiotherapy known to have activity against syngeneic melanoma in murine models is feasible and well tolerated in companion dogs with advanced stage, spontaneously arising melanoma or osteosarcoma and has immunomodulatory potential. Further studies evaluating the dose-dependent immunomodulatory effects of this immuno-radiotherapy combination are currently ongoing.

## Introduction

Metastatic cancer currently implies incurability for many patients and carries a guarded prognosis in humans and dogs alike, despite progress in available therapies. Cancer is a leading cause of mortality in adult companion dogs in North America [1], and strong genetic and molecular similarities and a shared environment make dogs a valuable preclinical model for human cancer [2]. Canine melanoma, in particular, represents a strong translational model for research due to its similarities to human mucosal melanoma and cutaneous triple negative melanoma [3–5]. The development of an experimental tyrosinase DNA vaccine in humans following data in companion dogs [6] exemplifies this point. Immunotherapy remains a promising treatment option for melanoma [7], along with other cancer types [8], and combined targeted and conventional therapies remain a promising research avenue [9, 10].

The interface between radiation therapy (RT) and immunotherapy has become an important area of investigation in the last decade [11]. The immunologic effects of radiation on tumor cells and the tumor microenvironment (TME) are current areas of intense investigation. Inflammatory cytokine signaling and immune cell recruitment following irradiation can create a temporary immunostimulatory environment, [12–14], with increased presentation of tumor-antigens that may serve as an *in situ* vaccine for immune recognition. Occasionally, these local immunomodulating effects can also result in abscopal responses that are also immune mediated [15–17]. The *in situ* vaccine effect may be further enhanced when combined with immunotherapy [15]. Combined, sub-ablative external beam radiation therapy (EBRT) and systemic and local immunotherapeutic strategies have been shown in some

settings to increase efficacy against primary tumors and distant metastatic sites [18–20] and novel combinations of radiation and immunotherapeutic strategies are being investigated to capitalize on this potential. A lack of demonstrated efficacy with RT alone and the need to improve on this with innovative approaches to better prime and propagate an *in situ* vaccine form the basis of our group's combination immune-radiotherapy approaches [21].

The immunocytokine (IC) fusion protein, hu14.18-IL2 that consists of human recombinant IL2 (hrIL2) fused to humanized anti-disialoganglioside (GD2) monoclonal antibody (mAb) has been evaluated in humans for the treatment of melanoma and neuroblastoma [22, 23]. As GD2 is a disialoganglioside, it is not species specific, and the 14.18 mAb can recognize GD2 on mouse, human and canine melanoma. Importantly, we and others have shown that canine melanoma and some soft tissue sarcomas express GD2 [24] (S1 and S2 Figs), and hrIL2 has significant immunostimulatory activity in dogs [25]. Further, the combination of hrIL2 and the mouse-human chimeric 14.18 anti-GD2 mAb used in this trial has been shown *in vitro* to induce an antibody-dependent cell mediated cytotoxicity (ADCC) response in canine melanoma cells (24). When hu14.18-IL2 is given intratumorally (IT) it results in increased activated T and NK cell infiltrates and tumor inhibition [26]. Further, in mouse models, combined EBRT and IT hu14.18-IL2 (IT-IC) create an *in situ* vaccine effect resulting in enhanced tumor response characterized by NK cell and CD8+ T cell infiltration, along with an antitumor memory T cell response [27].

The use of EBRT and IT-IC to create an *in situ* vaccine has potential limitations in the face of metastatic disease. Our group has shown in murine models that second non-irradiated tumors can serve as a nidus for immunosuppressive cells (e.g., Tregs), and these mediate systemic immunosuppressive effects that antagonize the EBRT/IT-IC generated *in situ* vaccination effect–a phenomenon referred to as concomitant immune tolerance (CIT) [28]. In our murine models, CIT is radiation sensitive since delivering moderate-dose (~ 12 Gy) RT to secondary tumor sites can overcome CIT and enable *in situ* vaccine regimens to destroy both primary and distant tumors [28]. Due to marked radiosensitivity of lymphocytes, doses of RT (~ 2–5 Gy) can also inhibit CIT (unpublished observations, Morris ZS). While it is not typically feasible to deliver EBRT to all sites of metastatic disease (due to immune suppression and inability to specifically target all microscopic disease), it may be possible to use targeted radionuclide therapy (TRT) to immunomodulate the TME of all tumor sites in the setting of metastatic disease. Indeed, simply applying whole-body low-dose EBRT (~ 2–5 Gy) does not have the same CIT abrogating effect in these models [29].

To overcome these limitations of EBRT, our group has investigated alkylphosphocholine (APCh) analogs that preferentially incorporate into cancer cell cytoplasmic membranes agnostic of species or tumor histology and can chelate a variety of diagnostic and therapeutic radionuclides [30]. NM600-radiometal chelates can thus be used to systemically deliver a targeted dose of radiation therapy preferentially to the collective tumor burden throughout the body while differentially sparing normal tissues. We have successfully used the theranostic pairing of $^{86}$Y-NM600 (for PET imaging and subject-specific dosimetry calculations) and $^{90}$Y-NM600 (for TRT) in mice to deliver TME modulating radiation therapy (~ 2–5 Gy) to second tumors, thus abrogating CIT [27, 28, 31]. This tri-modality immuno-radiotherapy approach (sub-ablative EBRT and IT-IC to the primary tumor to create an *in situ* vaccination and TRT to secondary tumors to abrogate CIT) theoretically could modulate the collective TME in a manner that promotes the propagation of an antitumor immune response to multifocal metastatic disease, as we have demonstrated by delivering EBRT to both primary and secondary tumor sites in murine models [28].

The overriding goal of the studies presented here were to determine the feasibility of translating this tri-modality immuno-radiotherapy approach into companion dogs with advanced

stage cancer by confirming tumor-selective uptake of NM600, performing subject-specific dosimetry of $^{90}$Y-NM600 and evaluating the adverse event (AE) profile of the combined immunotherapy protocol. Such an immunocompetent large animal surrogate with clinically relevant spontaneous-arising and heterogenous tumor/TME should better recapitulate the human condition, cross-validate our mouse data, and serve as an important translational bridge to human trials. Here we present data on companion dogs with advanced cancer treated with all components of our tri-modality approach, both individually and in combination.

## Materials and methods

### IACUC approvals

All procedures and treatments performed on laboratory and client-owned companion dogs were approved by the Institutional Animal Care and Use Committees of the University of Wisconsin-Madison School of Veterinary Medicine (Approvals V006037 and V006123). Written informed consent was also obtained from all companion dog caregivers prior to entry into trials. All companion dogs received their protocol treatments at the University of Wisconsin Veterinary Care (UWVC) hospital.

### Safety and assessment of Adverse Events (AE) in all treatment groups

AEs were observed, graded and attributed for all dogs in all single and combination treatment groups throughout the observation period using the Veterinary Cooperative Oncology Group —Common Terminology Criteria for Adverse Events (VCOG-CTCAE, v1.1) [32]. All dogs had to have a pretreatment constitutional clinical sign status of 0 or 1 (normal or asymptomatic/mild symptoms but able to function as an acceptable pet) according to VCOG-CTCAE v1.1 criteria at study entry. In all treatment groups, prior radiation therapy or immunotherapy was an exclusion criteria and a minimum 2-week washout from previous chemotherapy was required. Prior surgery was allowable as long as post-surgical recurrence met the 2 cm minimum longest diameter cut-off.

### Single-agent intratumoral immunocytokine (hu14.18-IL2) protocol

Client-owned companion dogs with confirmed neoplasia and an accessible primary tumor of at least 2 cm longest diameter were eligible for study. IT hu14.18-IL2 was given once daily over three consecutive days and was provided by Apeiron Biologics [22, 23]. A starting dose of 2mg/m$^2$/day was used, with dose-escalation performed in a 3x3 cohort design to a maximum intended dose (MID) of 12 mg/m$^2$/day. The MID was chosen from extrapolation of IT hu14.18-IL2 experience in mice [26] as well as representing the maximally tolerated dose for 3 consecutive days of intravenous hu14.18-IL2 in children with recurrent or refractory neuroblastoma and melanoma [33]. Prior to treatment, a baseline physical examination with tumor measurements was performed, blood was collected for clinical assessment of complete blood count (CBC) and biochemistry panel, and urine was collected for clinical urinalysis. Additional blood was also collected for banking of serum, plasma and peripheral blood mononuclear cells (PBMC) for subsequent *in vitro* analysis. Clinical tumor staging included aspirate of regional lymph nodes, thoracic radiographs and/or thoracic/abdominal CT as clinically indicated. A primary tumor biopsy was collected and divided between formalin-fixation, OCT and snap-frozen (OCT and snap-frozen samples subsequently stored at -80s°C) for future analysis (e.g., GD2 expression level, tumor infiltrating immune populations). The hu14.18-IL2 was provided as lyophilized vials (4 mg/vial; each ml prior to lyophilization contained 4 mg/ml hu14.18-IL2, 2% sucrose, 80mM L-arginine, 10mM citric acid, 0.2% polysorbate 20, pH 5.5) and was

reconstituted with either 0.5 ml or 0.25 ml 0.9% (w/v) NaCl for injection at a concentration of either 8 mg/ml or 16 mg/ml. The volume chosen in each case was dependent on the size of the primary tumor to be injected. The hu14.18-IL2 was injected (approximate rate of 0.5 mL over ~3 minutes) in sedated or anesthetized dogs through a 25 gauge needle, with additional brief breaks allowed during administration, if needed, for puncture site clotting to minimize leakage of injected material. The total volume was administered using multiple needle re-directions into the target lesion to distribute the injected material as uniformly as possible within and surrounding the lesion. Injections were performed daily for 3 consecutive days. Blood and urine collections were repeated on Day 10 and Day 24 post-treatment. Tumor biopsy was repeated on Day 10, 17 and 24 post-treatment and processed as above.

## EBRT and IT-IC combination protocol

Following completion of single agent IT-IC dose escalation safety, to bridge with our planned EBRT/TRT/IT-IC tri-modal protocol (see below), a safety evaluation using an EBRT/IT-IC combination treatment protocol was performed in 3 companion dogs with histologically confirmed malignant melanoma. All dogs had an accessible primary lesion of $\geq$ 2 cm in size and all underwent similar pretreatment screening, staging and biospecimen collections as previously outlined for IT-IC single-agent studies. EBRT was delivered by image-guided intensity modulated radiation using helical tomotherapy (TomoTherapy HiArt Treatment System[R], Accuray Inc., Sunnyvale, CA, USA). Patients underwent CT simulation using a maxillary immobilization system and vacuum formable mattress [34]. Axial CT slice thickness of 1.25–2.5 mm was used to image the entire treatment region. The gross tumor volume (GTV) was defined based on the CT imaging by the attending veterinary radiation oncologist. A clinical target volume (CTV) of 10–20 mm was applied to the GTV and confined by anatomic tissue planes considered by the clinician to limit spread of the tumor. A 2 mm planning target volume (PTV) was applied to the CTV. Clinically involved lymph node volumes were delineated as lymph node GTV and an 8mm isotropic lymph node PTV was applied. Normal tissues considered at risk were contoured and plans optimized to minimize dose to these structures while prioritizing tumor target coverage. These 3 companion dogs represent the first in an ongoing, accruing randomized trial comparing two EBRT fractionation arms. In one arm, a single fraction of 8 Gy is delivered to 95% of the PTV. In the other arm, 3 fractions of 8 Gy are delivered to > 95% of the GTV, with 95% of the PTV receiving 6 Gy, on a Monday-Wednesday-Friday schedule. Five days following the single fraction or the last of 3 fractions, three consecutive days of IT-IC injections were initiated (12 mg/m$^2$) as outlined above. Post treatment AE characterizations and biospecimen collections schedules are outlined in Fig 1.

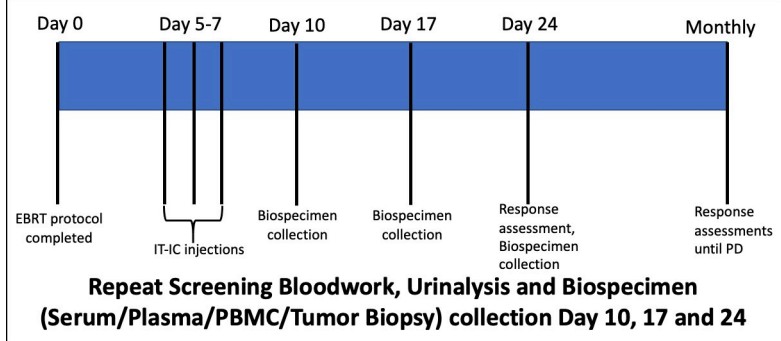

**Fig 1. Protocol schema for tumor-bearing companion dogs receiving external beam radiation therapy in combination with intratumoral hu14.18-IL2 immunocytokine.**

## NM600 safety evaluation

2-(trimethylammonio)ethyl(18-(4-(2-(4,7,10-tris(carboxymethyl)-1,4,7,10-tetraazacyclodode-can-1-yl)acetamido)phenyl)octadecyl) phosphate (NM600) was kindly provided by Archeus Technologies (Madison, WI). As a prelude to our investigation of the theranostic pairing of [86]Y-NM600/[90]Y-NM600 for TRT, four purpose-bred laboratory adult beagles (two neutered male and two spayed females) were included in an acute single-dose target animal safety study of NM600. All dogs were acclimated to research housing and then underwent initial baseline physical examination along with baseline collection of CBC, biochemistry panel and urinalysis. Each dog was given 2 mg of NM600 IV on Day 0. This dose is $\geq$ 10-fold the NM600 dose used when chelated with [90]Y radionuclide for TRT therapy in tumor-bearing companion dogs (i.e., 10 μg NM600/mCi [90]Y delivered). Dogs were assessed daily for clinical signs of toxicity (i.e., temperature, pulse, respiratory rate, body weight, lethargy, etc) and clinicopathologic (i.e., CBC, biochemistry panel and urinalysis) AEs were evaluated on days 0, 7 and 21 post-NM600 injection.

## EBRT, [86]Y-NM600/[90]Y-NM600 theranostic TRT and IT-IC combination protocol [86/90]Y production and radiochemistry

[86]Y (β+, tR1/2R = 14.7 h) was produced in a GE PETrace (GE Healthcare, Waukesha, WI) bio-medical cyclotron via irradiation of enriched [[86]Sr]SrCO$_3$ (96.4 ± 0.1%) targets with 16.4 MeV protons as described previously [35]. [90]Y was purchased from Perkin Elmer as [90]YCl$_3$. Radiola-beling of NM600 with [86/90]Y and purification was accomplished as previously described [31]. Briefly, 5–20 mCi of 90Y or 86Y was buffered with 0.1 M NaAOC (pH = 5.5) and 10 ug/mCi of NM600 were added. The reaction was incubated at 95°C for 30–60 min under constant shaking (500 rpm). Radiolabeled [86/90]Y-NM600 was then purified by solid-phase extraction chromatography, and the activity was reconstituted in 5–10 mL of normal saline containing 0.1% v/v Tween 20 at an approximate activity concentration of 2 mCi/ml. The formulation was then filtered through a 0.2 um filter into a sterile sealed vial.

**Companion dog therapy protocol.** Five client-owned companion dogs with advanced stage cancer were recruited with intent to treat each with combined EBRT, [86/90]Y-NM600 ther-anostic TRT and IT-IC according to the schedule in Fig 2. Inclusion criteria included the presence of a measurable index lesion accessible to biopsy with at least one distant metastatic lesion. All patients had baseline and treatment cycle physical examinations, CBC, biochemistry

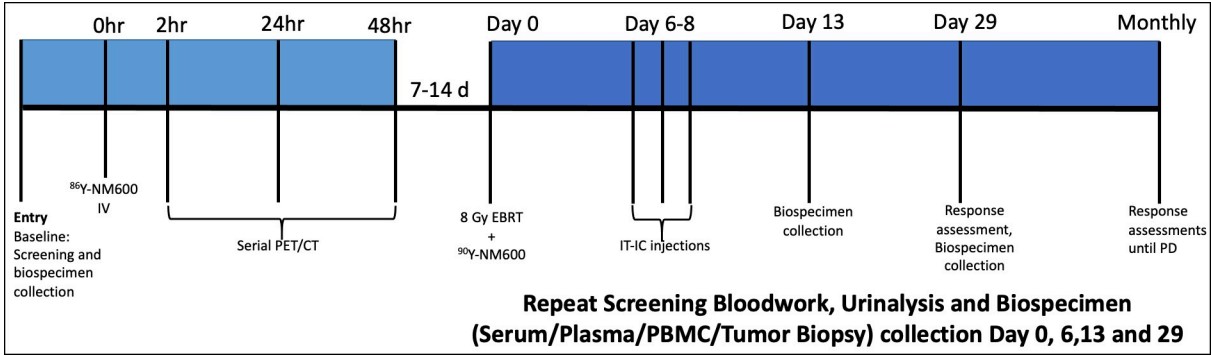

**Fig 2. Protocol schema for tumor-bearing companion dogs receiving external beam radiation therapy in combination with intratumoral hu14.18-IL2 immunocytokine and [86/90]Y-NM600 targeted radionuclide therapy.**

panels, urinalyses performed along with blood and tissue biospecimen collections for planned *in vitro* immune assays as outlined in Fig 2.

Following enrollment each patient received 5–10 mCi [86]Y-NM600 IV bolus injection over 5 minutes followed by serial PET-CT imaging at 1–2, 24, and 48 hours post-injection on a 64-slice scanner (GE Discovery MI, GE Healthcare, Waukesha, WI). Radiopharmaceutical Assessment Platform for Internal Dosimetry (RAPID), a state-of-the-art Monte Carlo-based platform was then used to determine the radiation dose that would be delivered over a given time to tumor(s) and normal tissues following injection of a given activity of a given radionuclide [36–38], resulting in a personalized treatment plan. We have demonstrated the ability to use RAPID to accurately predict the subject-specific dosimetry of [90]Y-NM600 from [86]Y-NM600 PET/CT imaging in murine tumors and the capacity of [90]Y-NM600 to deliver an immunomodulatory radiation dose in these settings [31, 39]. Monte-Carlo calculations were used to determine an [90]Y-NM600 prescription for each dog that would deliver a mean tumor dose > 2 Gy to all systemic (metastatic) sites and a mean bone marrow dose ~ 1 Gy or less. The lumbar vertebral bone marrow compartment was consistently the region of the bone marrow with the greatest [86]Y-NM600 uptake and was therefore used for bone marrow dosimetry calculations. If a ≥ 2 metastatic tumor:bone marrow dose was not achievable based on RAPID analysis, the dog was deemed ineligible to receive to [90]Y-NM600 TRT. As the primary tumor receives 8 Gy EBRT, the calculated TRT dose to the primary was not considered relevant to the 2:1 dose cut off for TRT dose eligibility.

One or 2 weeks following [86]Y-NM600 imaging and dosimetry, dogs were treated with 8 Gy EBRT to their primary/index tumor followed on the same day by the prescribed [90]Y-NM600 IV injection. EBRT was applied as described previously with the exception that GTV, rather than CTV was contoured as the intent in dogs with metastatic disease was not to necessarily target peritumoral microscopic disease, rather to create an *in situ* vaccine effect. Following [86]Y-NM600 and [90]Y-NM600 injections, all dogs were hospitalized in a UW-Madison radiation safety approved holding area at UWVC and radiation exposure at 1 m (mR/hr) measured daily until they met the radiation safety tolerance for release to the companion owner (< 1mR/hr at 1 m).

IT-IC injections were subsequently given over three consecutive days to the primary/index tumor site at 6 mg/m$^2$/d, beginning 6 days following the EBRT/TRT treatment as previously outlined. The 6 mg/m$^2$/d IT-IC dose was used as this ongoing TRT dose escalation study was run contemporaneously with the IT-IC single-agent trial and was initiated after the 6 mg/m$^2$/d cohort was completed but before the 12 mg/m$^2$/d cohort was completed.

## Exploratory analysis of immunomodulatory effects

Biospecimens collected at serial time points (Fig 2) from the 3 dogs with malignant melanoma completing the trimodal (EBRT/IT-IC/TRT) immuno-radiotherapy protocol were interrogated for immunomodulatory changes resulting from therapy. These included primary tumor gene expression analysis, primary tumor infiltrating lymphocyte (TIL) assessment (both by immunohistochemistry [IHC] and Nanostring® Cell Type Profiling) and flow-cytometric lymphocyte subset analysis of peripheral blood mononuclear cells (PBMC). Additionally, in one patient undergoing warm necropsy at the time of euthanasia, biospecimens (primary and metastatic tumors, PBMC) were also included in these analyses.

**Gene expression analysis.** Tumor specimens were collected and immediately snap-frozen in liquid nitrogen before being stored at -80˚C until time of analysis. RNA from frozen tissue specimens were extracted using Maxwell 16 LEV simplyRNA tissue kit, AS1280 (Promega) following manufacturer's protocol. Briefly, samples were homogenized in a Maxwell proprietary

solution consisting of homogenization solution and 1-Thioglycerol. Lysis buffer was added and samples run on Maxwell (Promega). Then, total canine RNA was used with nCounter® Canine Immuno-Oncology (IO) Panel kit, XT-CSPS-CIO-12 (NanoString Technologies, Inc., Seattle, WA; https://www.nanostring.com/products/ncounter-assays-panels/oncology/canine-io/) to measure gene expression of 800 genes across 47 annotated pathways following manufacturer's protocol. Briefly, the NanoString® nCounter® system uses unique color-coded probes that act as a barcode to identify and count individual transcripts without reverse transcription or amplification. The platform performs similarly to real-time PCR (R2 = 0.95) and is more sensitive than microarrays [40, 41]. Samples were first setup in the nCounter Prep® Station to clean and process the samples. Then, samples were transferred to the nCounter® Digital Analyzer, a multi-channel epifluorescence scanner to acquire the data. Digital data was transferred to ROSALIND® (San Diego, CA; https://rosalind.onramp.bio/) a cloud-based software suite with a HyperScale architecture, for QC, data analysis and visualization. Read Distribution percentages, violin plots, identity heatmaps, and sample MDS plots were generated as part of the QC step. Normalization, fold changes and p-values were calculated using criteria provided by NanoString®. ROSALIND® follows the nCounter® Advanced Analysis protocol of dividing counts within a lane by the geometric mean of the normalizer probes from the same lane. Housekeeping probes to be used for normalization are selected based on the geNorm algorithm as implemented in the NormqPCR R library [42]. Fold changes and pValues are calculated using the fast method as described in the nCounter® Advanced Analysis 2.0 User Manual. P-value adjustment is performed using the Benjamini-Hochberg method of estimating false discovery rates (FDR). Clustering of genes for the final heatmap of differentially expressed genes was done using the PAM (Partitioning Around Medoids) method using the fpc R library [43] that takes into consideration the direction and type of all signals on a pathway, the position, role and type of every gene, etc. Hypergeometric distribution was used to analyze the enrichment of pathways, gene ontology, domain structure, and other ontologies. The topGO R library [44], was used to determine local similarities and dependencies between GO terms in order to perform Elim pruning correction. Several database sources were referenced for enrichment analysis, including Interpro [45], NCBI [46], MSigDB [47, 48], REACTOME [49], WikiPathways [50]. Enrichment was calculated relative to a set of background genes relevant for the experiment.

Abundance of various cell populations is calculated on ROSALIND® using the NanoString® Cell Type Profiling Module. ROSALIND® performs a filtering of Cell Type Profiling to include results that have scores with a p-Value greater than or equal to 0.05. NanoString Cell type profiling included in the Canine IO panel uses gene co-expression signatures to determine the relative abundance of immune cell types within a sample [51].

**Tumor Infiltrating Lymphocyte (TIL) immunohistochemistry.** Immunohistochemistry was performed on 4 micron formalin-fixed, paraffin-embedded tumor sections using a Ventana Discovery Ultra instrument (Roche, Tucson, AZ). Deparaffinization was carried out on the instrument, as was heat-induced epitope retrieval in the form of "cell conditioning" with CC1 buffer (Ventana #950–500), an EDTA based buffer, for approximately 32 minutes at 95˚C (recommended). The primary antibodies used and the conditions for each of CD3, CD4, CD8 and FOXP3 are listed in S1 Table. Both canine normal lymph node and human tonsil controls were used for each condition. All IHC slides were read by a single pathologist (Gasper) and read out as the sum of the total lymphocyte distribution that occupied the tumor tissue with the lymphocyte density found in ten 400X fields.

**Flow cytometric analysis of peripheral circulating lymphocyte subsets.** Cryopreserved PBMC (pretreatment, day 6 and day 29+/-3d) were briefly thawed in a 37˚C water bath and washed in minimal essential medium containing 10% fetal bovine serum (MEM10, Corning).

Samples were washed again in PBS and stained with a viability dye (Ghost Red 780, Tonbo Biosciences, San Diego. CA) for 45 minutes in a total volume of 1 mL. Samples were then washed in staining buffer consisting of phosphate buffered saline (PBS, Corning, Corning, NY) with 3% FBS (Corning, Corning, NY) and 1mM EDTA (Gibco, Life Technologies, CA) before staining with a cocktail of surface antibodies diluted in staining buffer for 1 hour. After washing with staining buffer again, cells were incubated in 1 mL of fixation/permeabilization buffer (Life Technologies) for 30 minutes. Permeabilization buffer was then used to wash the cells before a 1-hour incubation with the FoxP3 intracellular antibody diluted in permeabilization buffer. Intracellular staining was performed using FoxP3 Transcription Factor Staining Kit (Life Technologies, Carlsbad, CA) following the manufactures protocol. Details on all antibodies used in this study are listed in S2 Table. Finally, all samples were washed again in permeabilization buffer and resuspended in staining buffer while awaiting flow cytometer acquisition. All antibody staining steps were performed in a final volume of 100 μl. All incubation steps were performed at 4°C.

Controls included unstained cells and fluorescent minus one (FMO) for proper gating and recognition of positive populations. Flow cytometry acquisition was performed on a Thermo Fisher Attune NxT acoustic focusing flow cytometer, utilizing the blue (488nm), yellow/green (561nm), violet (405nm) and red (633nm) lasers. Compensation settings were conducted using UltraComp eBeads (Invitrogen, Carlsbad, CA). Flow cytometry data was analyzed using FlowJo software (BD Biosciences, Franklin Lakes, NJ). Lymphocytes were identified based on cell characteristic properties in the forward (FSC) and side (SSC) scatter. The frequency of $CD8^+$ and $CD4^+$ peripherally circulating lymphocytes were defined as the proportion of $CD8^+$ or $CD4^+$ cells within the $CD3^+$ cell population, respectively. The frequency of regulatory T lymphocytes (Tregs) was calculated based on percentage of $CD3^+/CD4^+$ lymphocytes that were $CD25^+$ and $FoxP3^+$. The frequency of NK cells was defined as the percentage of $CD3^-$ cells that were $CD5^{low/dim}$. An example of the gating strategy used is presented in S3 Fig. The results were expressed as relative proportions of the phenotypic subsets measured using 4-quadrant analysis and histogram with the FlowJo software.

**Assessment of GD2 expression by immunofluorescence microscopy.** Fresh tissue was embedded and frozen in OCT (Fisher HealthCare, Houston, TX) and two 10 um sections adhered to slides which were fixed with -20°C acetone for 10 min and washed in tap water for 10 min to remove residual OCT. Sections were incubated with 10% FBS in PBS for 1 hr at room temperature and washed with 1% FBS in PBS. A GD2 expressing B78 mouse tumor was used as positive control. The top section on each slide received no anti-GD2-PE and the bottom section was incubated with 1:100 anti-GD2-PE (BioLegend clone 14.G2a, catalogue number 357304) in 1% FBS overnight at 4°C in a dark humidity chamber Slides were then washed in 1% FBS in PBS for 5 min, then washed twice in PBS for 5 min. Slides were counterstained with 1 drop of DAPI during one of the washing steps. Slides were then washed with deionized water 3 times for 5 min each and coverslipped. Multiple fields of view were imaged on EVOS M5000 (ThermoFischer), using RFP and DAPI cubes as well as transparent light at 20X magnification. All images were recorded with the same setting for RFP cube by using the positive and negative control (0.0086 on EVOS for light). S2 Fig presents immunofluorescence images of a GD2 expressing canine melanoma and a canine soft tissue sarcoma assessed with this methodology.

**Statistical analysis.** As a proof-of-concept safety and feasibility study, results were largely descriptive. Continuous variables were summarized using means and ranges. Cell type profiling and PBMC flow-cytometric data analysis and graphs were generated and non-parametric repeated-measures ANOVA (Friedman test) with Dunn's multiple comparisons test were performed using Prism 9, version 9.0.2, software (GraphPad).

## Results

### Single-agent IT-IC is safe and feasible in tumor-bearing companion dogs

Eight dogs were accrued solely for this protocol cohort (three in the 2 mg/m$^2$/d cohort, two in the 6 mg/m$^2$/d cohort and three in the 12 mg/m$^2$/d cohort); additionally, AE and safety data from 1 dog who received IT-IC (6 mg/m$^2$/d) in combination with EBRT in a contemporaneous study was included in the decision to move from the 6 to the 12 mg/m$^2$/d cohort. Treated dogs consisted of three Golden Retrievers, and one each of Doberman Pinscher, Greyhound, Springer Spaniel, Border Collie and mixed breed. The group consisted of four spayed females, three neutered male dogs and one intact male, and had a median age of 9.5 years (3–15 years) and median weight of 31.1kg (15.2–39.2 kg). Five dogs had histologically confirmed oral melanoma, two osteosarcomas, and one had an oral growth initially diagnosed as melanoma during screening but was subsequently reclassified on histologic review as benign proliferative glossitis after therapy was completed. All dogs completed the planned series of three IT-IC injections.

The AE profile for IT-IC consisted of grade 1 and 2 events, summarized in Table 1. One dog in the 2 mg/m$^2$ cohort exhibited self-mutilation of the tumor site 48 hours following injection, and transient pain on injection was noted in several dogs depending on anatomic site and degree of sedation employed for injection. All AEs noted were transient, and no dose-limiting AEs occurred.

While antitumor outcome was not a primary endpoint, all dogs, with the exception of the subject with a reclassified benign tumor, were determined to have progressive disease (PD) at day 21 post treatment based on VCOG-RECIST criteria [52].

### EBRT and IT-IC combination therapy is safe and feasible in tumor-bearing companion dogs

Dog demographics consisted of two neutered males and one spayed female, with age and weight ranging from 7–10 years and 6.6.-43.8 kg, respectively. Three breeds were represented; Labrador Retriever, Bernese Mountain Dog and one mixed breed. All three had histologically confirmed advanced stage III or IV melanoma according to WHO TNM classification of canine melanoma [53]; one each of T2N0M1, T2N2M0 and T2N1M0. Two dogs were randomized to receive a single 8 Gy fraction and a third dog to three 8 Gy fractions as outlined in methods.

All dogs completed the prescribed EBRT and IT-IC treatments, and no dose-limiting AEs were observed. The AE profile for this combination was similar to that described for IT-IC alone; all were transient, low-grade (1 or 2) and are summarized in Table 1.

While antitumor outcome was not a primary endpoint, the first dog randomized (single fraction group) experienced a partial response (PR, 67% reduction in longest diameter) of the primary tumor at the 24 day post-treatment reevaluation. Thoracic radiographs, however, revealed progressive growth of the pulmonary lesion at the 1-month reevaluation with continued PR of the primary. To ensure the pulmonary lesion was indeed PD and not immunologic pseudoprogression, repeat thoracic radiographs were performed 2 and 4 months later and the pulmonary nodule was stable. The second dog in this cohort experienced SD for 2 months post-EBRT/IT-IC, but PD was documented at the 3-month reevaluation, both in primary and nodal sites. The final dog in this cohort had SD at 1 month and PD documented at the 2-month reevaluation.

### NM600 is safe in normal laboratory dogs

No dogs developed clinically significant physical examination changes (e.g., body weight, body temperature, pulse and respiratory rate) over the 21-day follow-up period. In only two

**Table 1. Adverse event summary for all dogs by treatment group.**

| Treatment Group | IT-IC alone | | | EBRT/ IT-IC | EBRT/IT-IC and TRT |
|---|---|---|---|---|---|
| Adverse Event | Cohort 1 (2mg/m², n = 3) | Cohort 2 (6mg/m², n = 2) | Cohort 3 (12mg/m², n = 3) | (12 mg/m² IT-IC; n = 3) | (6 mg/m² IT-IC; n = 4) |
| **Anemia** | | | | | |
| Grade 1 | | | | | 1 |
| **Eosinophilia** | | | | | |
| Grade 1 | | | 2 | 1 | |
| **Thrombocytopenia** | | | | | |
| Grade 1 | 1 | 1 | | | 1 |
| **Neutropenia** | | | | | |
| Grade 1 | | | | | 1 |
| **Lymphopenia** | | | | | |
| Grade 1 | | | | | 2 |
| **Lymphocytosis** | | | | | |
| Grade 1 | | | 1 | | |
| **Monocytosis** | | | | | |
| Grade 1 | | | 1 | | 2 |
| **ALKP elevation** | | | | | |
| Grade 1 | | | 1 | | 2 |
| Grade 2 | | | | 1 | 1 |
| **ALT elevation** | | | | | |
| Grade 1 | | | | | 1 |
| **CK elevation** | | | | | |
| Grade 1 | | | 1 | | 1 |
| **Hyperalbuminemia** | | | | | |
| Grade 1 | | | | | |
| **Hyperglobinemia** | | | | | |
| Grade 1 | | | | 1 | |
| **Hypertryglyceridemia** | | | | | |
| Grade 1 | | | | | 1 |
| **Hyponatremia** | | | | | |
| Grade 1 | | | | 1 | 1 |
| **Hypocalcemia** | | | | | |
| Grade 1 | | | | | 2 |
| **Hypomagnesemia** | | | | 1 | |
| Grade 1 | | | | | |
| **Hypermagnesemia** | | | | | |
| Grade 1 | | | | | 1 |
| **Hematuria** | | | | | |
| Grade 1 | | | | 2 | |
| **Hyporexia** | | | | | |
| Grade 1 | | 2 | | | |
| Grade 2 | | | | | 2 |
| **Nausea** | | | | | |
| Grade 1 | | | | | 1 |
| **Diarrhea** | | | | | |
| Grade 1 | 1 | | | 2 | 1 |
| Grade 2 | | | | | 2 |
| **Lethargy/Fatigue** | | | | | |

*(Continued)*

**Table 1.** (Continued)

| Treatment Group | IT-IC alone | | | EBRT/ IT-IC | EBRT/IT-IC and TRT |
|---|---|---|---|---|---|
| Adverse Event | Cohort 1 (2mg/m², n = 3) | Cohort 2 (6mg/m², n = 2) | Cohort 3 (12mg/m², n = 3) | (12 mg/m² IT-IC; n = 3) | (6 mg/m² IT-IC; n = 4) |
| Grade 1 | | | 1 | | |
| Pruritis | | | | | |
| Grade 2 | | | | 1 | |
| Hypersensitivity | | | | | |
| Grade 2 | | | | | 1 |

ALKP = alkaline phosphatase, ALT = alanine aminotransferase, CK = creatinine kinase.

instances did a dog have a recorded laboratory value outside the normal range not present at pretreatment baseline; one dog developed a grade 2 hypoglycemic event (47 mg/dL) at day 21 post-treatment, and another developed a grade 1 hyperglycemic event (134 mg/dL) at day 21 post-treatment. Low grade fluctuations in blood glucose levels in these dogs was likely attributable to a lack of standardization in feeding schedule and time from blood collection to serum separation. All dogs maintained urine concentrating ability; however, 3 dogs had 1 incident each of grade 1 microscopic hematuria attributed to urinary catheterization for sample collection.

## EBRT, IT-IC and $^{86}$Y-NM600/$^{90}$Y-NM600 theranostic TRT is safe and feasible in tumor-bearing companion dogs

All dogs had advanced stage IV disease and signalment and tumor characteristics are summarized in Table 2. Following $^{86}$Y-NM600 PET/CT imaging, all dogs exhibited differential tumor/normal tissue uptake (Figs 3 and 4). Four of 5 dogs had differential tumor to bone marrow uptake affording the ≥2 safety margin in tumor:bone marrow absorbed dose to allow continuation to $^{90}$Y-NM600 TRT treatment and each dogs' prescription is listed in Table 2. The one dog who did not exhibit adequate differential $^{86}$Y-NM600 uptake was removed from trial and received EBRT/IT-IC combination therapy on a compassionate use basis. The remaining four dogs all received the combined treatment approach as planned. Protocol deviations included one dog who received $^{90}$Y-NM600 on Day 1 instead of Day 0 concurrently with EBRT, and one dog (Dog 3) who received a second $^{90}$Y-NM600 treatment two weeks after the first.

**Table 2. Patient characteristics of dogs receiving EBRT/TRT/IT-IC combination protocol.**

| ID | Breed | Sex | Age | Weight | Primary/Index Tumor | Distant Lesions | Prescribed $^{90}$Y-NM600 |
|---|---|---|---|---|---|---|---|
| 1 | Cane Corso | MN | 5 yr | 44.6 kg | Metastatic Osteosarcoma* (TxN0M1) | Pulmonary nodules, left proximal humerus | 17.55 mCi |
| 2 | Cocker Spaniel | MN | 9 yr | 18.2 kg | Oral Melanoma (T2N1M1) | Pulmonary nodules, mandibular lymph node | 8.49 mCi, 9.02mCi‡ |
| 3 | Miniature Schnauzer | FS | 7 yr | 8.4 kg | Oral Melanoma (T1N1M1) | Regional lymph node, pulmonary nodule | 6.84 mCi |
| 4 | Mixed Breed | FS | 10 yr | 19 kg | Metastatic subungual melanoma# (TxN2M1) | Pulmonary mass, Jejunal mass | 9.64 mCi |

MN (Male Neutered), FS (Female Spayed).

*Primary tumor previously amputated, therefore the largest metastatic lesion (intramuscular, right proximal hip) was used as the index tumor receiving EBRT/IT-IC.

‡This dog received two cycles of $^{90}$Y-NM600, 2 weeks apart.

#The subungual primary tumor was previously excised, therefore the right epaxial muscle metastatic mass was used as the index tumor receiving EBRT/IT-IC.

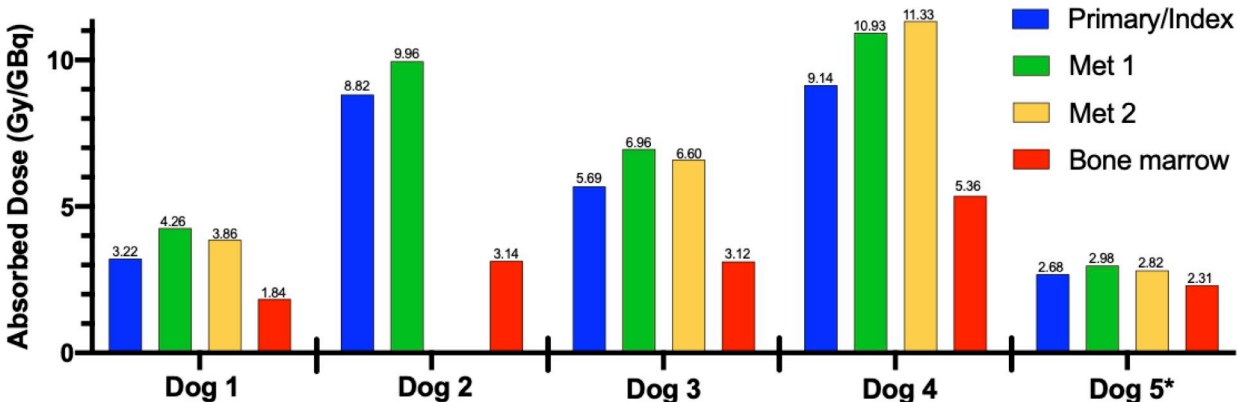

**Fig 3. Tissue absorbed dose (Gy/GBq) estimates for $^{90}$Y-NM600 using serial $^{86}$Y-NM600 PET/CT scans and the RAPID Monte Carlo-based analysis platform as described in the Methods section.** These estimates were used to determine the $^{90}$Y-NM600 prescription to achieve at least a 2 Gy absorbed dose to all metastatic lesions. Note that four (1–4) of 5 dogs met the a priori requirement for at least a 2:1 metastatic tumor:bone marrow differential uptake ratio. Dog 5 did not demonstrate the 2:1 requirement and was not subsequently treated with $^{90}$Y-NM600 targeted radionuclide therapy.

AEs were of low grade (1–2) and are presented in Table 1. One dog experienced a grade 2 hypersensitivity reaction immediately following $^{90}$Y-NM600 which was treated with a single diphenhydramine injection.

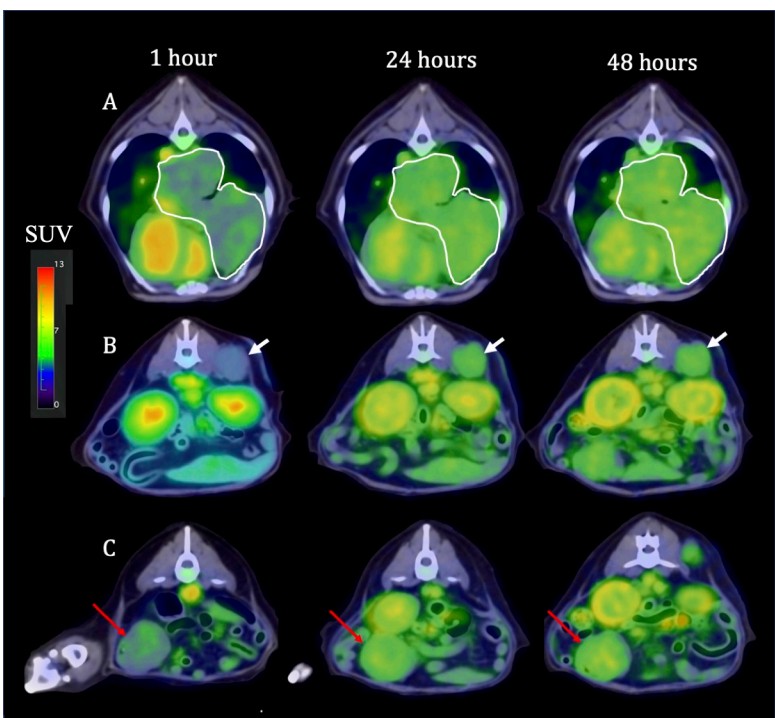

**Fig 4. Tumor selective uptake of $^{86}$Y-NM600 as documented by serial PET/CT in one illustrative case (Dog 4).** 1, 24 and 48 hour axial images at the level of (A) right middle and caudal lung lobe melanoma metastasis (outlined in white); (B) right epaxial muscle melanoma metastasis (white arrow) that was used as the index tumor for external beam radiation therapy and intratumoral hu14.18-IL2 immunocytokine to create an *in situ* vaccination; (C) jejunal melanoma metastasis (red arrow). Note the $^{86}$Y-NM600 is primarily in the vascular compartment at the 1 hour time point and then selectively accumulates into all metastatic sites at subsequent time points.

Three dogs died during the study follow-up period and all had a complete necropsy performed. Dog 1 experienced grade 5 cluster seizures and acute death while in home care at day 19 post-treatment; osteosarcoma metastasis to the right cerebral cortex was confirmed on necropsy. Dog 2 experienced a grade 3 seizure event followed by grade 3 obtundation and motor deficits at day 11. MRI confirmed the presence of an intra-axial lesion within the hippocampus. While transient improvement was noted with medical management, this patient eventually progressed and was euthanized at day 53 following therapy, and the intracranial mass was confirmed at necropsy as metastatic melanoma. Dog 3 is currently alive 13 months after completion of treatment; progressive pulmonary nodules were documented at the 5 month reevaluation and the subject was subsequently enrolled in an unrelated clinical trial. Dog 4 developed sudden grade 3 dyspnea, tachypnea and hypoxia resulting in euthanasia at day 25. Necropsy revealed fibrinous septic peritonitis secondary to rupture of the jejunal metastatic mass and a newly identified cardiac mass in the right atrium; both confirmed as metastatic melanoma.

### Exploratory analysis of immunomodulatory effects

Gene expression analysis was performed on 12 samples constituting the 3 dogs (dog 2, 3, and 4) with advanced stage melanoma receiving the trimodal (EBRT/TRT/IC-IT) immuno-radiotherapy protocol. For each dog, 3 serial index tumor samples (pretreatment, 6 d and 13d post initiation of treatment) were interrogated. For dog 4, 3 additional samples (index tumor and two distinct metastatic tumors) at necropsy (day 28 post treatment) were included in Cell Type Profiling (see below). RNA quality was excellent in 11/12 samples (S3 Table). Quality control data are presented in supplemental data (S4–S7 Figs). At the 6 day post-treatment timepoint, of the 800 genes interrogated, 65 were significantly over- and 4 under-expressed and these changes clustered into 8 of 47 annotated pathways within the Canine IO Panel (Fig 5). At the 13 day timepoint, 38 genes were significantly over and 1gene under expressed and these changes clustered into 15 annotated pathways within the Canine IO Panel (Fig 6). Six annotated pathways (costimulatory signaling, TNF superfamily, cytokines, interleukins, B cell function and NK cell function) were upregulated at both timepoints.

Using the Immune Cell Profiling Feature of nCounter, the relative abundance of several immune cell types were interrogated and are summarized in Fig 7. Numerically, T, CD8, CD45, TH1, and NK CD56$^{dim}$ gene signatures increased in relative abundance at 6 days following initiation of therapy and the latter two achieved statistical significance. While the relative abundance of these cell types were trending downward by day 13, they still remained higher than at pretreatment. Interestingly, in dog 4 where cell profiling was also performed on day 28 necropsy samples, while the relative abundance of these cell types remained high in the primary tumor, both metastatic lesions were similar to pretreatment primary tumor levels (Fig 8).

TIL assessment by IHC was only evaluable in 2 dogs, as the degree of melanin pigmentation in dog 4 was too abundant to differentiate staining patterns. Further, the CD4 mAb reactivity was too non-specific to be reliable for characterizing CD4 lymphocytes. Numerically, CD3, CD8 and FoxP3 TILs increased over the course of the treatment and was maximal at 13 days; however, these changes did not achieve significance (Fig 9).

Changes in peripheral circulating lymphocyte subsets during therapy are summarized in Table 3 and Fig 10. A representative example of the gating strategy for flow cytometric analysis is presented in supplemental S3 Fig.

Of the 3 dogs with melanoma entered in the trimodal immuno-radiotherapy protocol, one dog primary tumor (dog 2) was positive for GD2 expression by immunofluorescence microscopy.

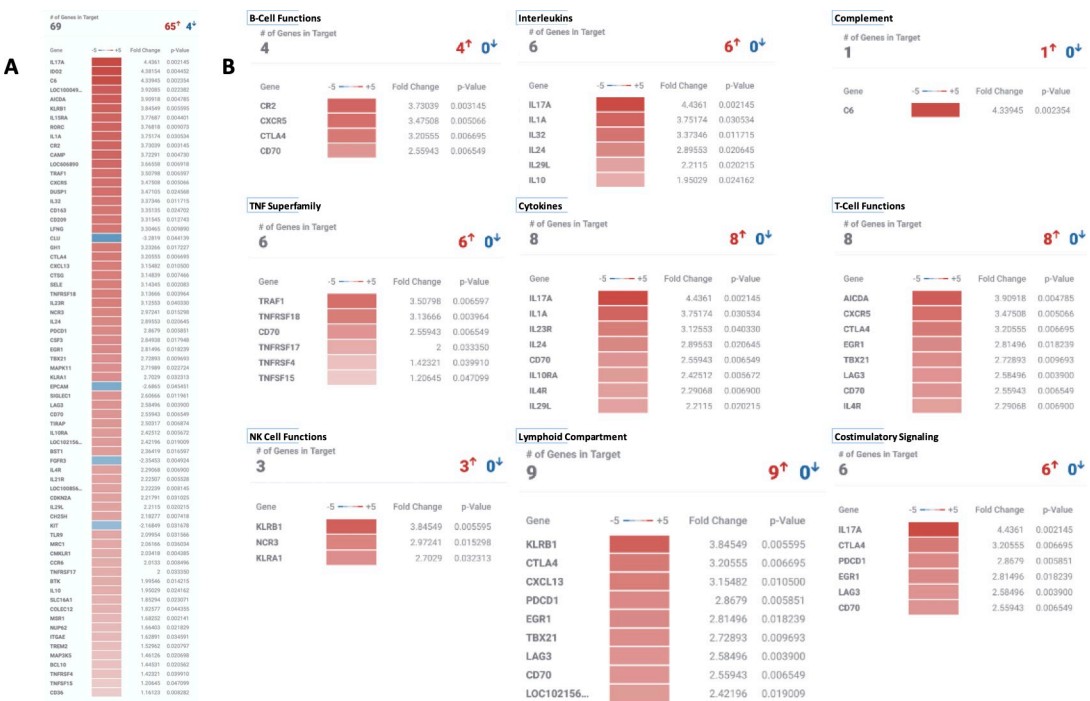

**Fig 5. Differential canine immuno-oncology gene expression at day 6 post initiation of therapy.** (A) Overall, 65 genes had significantly increased expression and 4 genes had significantly decreased expression compared to pretreatment samples. (B) Clustering of differential gene expression changes by biologically relevant immuno-oncology pathways.

## Discussion

Companion dogs with spontaneously arising tumors are compelling comparative models to human cancers for several reasons [2]: 1) They develop naturally occurring cancers with strong genetic and molecular similarities to human tumors. 2) Canine tumors develop in the setting of immune competence and undergo selective immune editing similar to human cancers. 3) The inter-individual and intratumor heterogeneity of canine cancers, which occur across a spectrum of age, sex, and breeds, mimics the heterogeneity of human cancers. 4) Companion canine tumors develop in a native autochthonous microenvironment, often following environmental carcinogen exposures that are shared in common with humans. 5) Finally, and particularly germane to this study, the physical size and spatial distribution of tumors in companion dogs more closely mimic that in humans with cancer. This spatial similarity is critical for studying the interaction of TRT with the TME and lymphoid organs-at-risk (bone marrow, spleen, thymus, draining lymphatics). Because of these considerations, TRT dosimetry calculations using dogs should be more reliable for extrapolation to humans than mouse models.

Taken in totality, all aspects of the tri-modality combination immuno-radiotherapy protocol (EBRT/TRT/IT-IC) were well tolerated in tumor-bearing dogs, either when given alone or in combination. All AEs were transient, of low grade (1 or 2) and no dose-limiting toxicities were encountered (Table 1). As bone marrow toxicity would be the anticipated dose-limiting AE from TRT, it is of note that only a single transient grade 1 neutropenia and two transient grade 1 lymphopenias were observed. Therefore, our ≥2 metastatic tumor:bone marrow prerequisite when prescribing a 2 Gy $^{90}$Y-NM600 TRT dose to metastatic lesions appears safe in tumor-bearing companion dogs and likely can be escalated.

Low grade fluctuations in blood glucose levels in normal laboratory beagles receiving single-agent NM600 were likely attributable to a lack of standardization in feeding schedule as

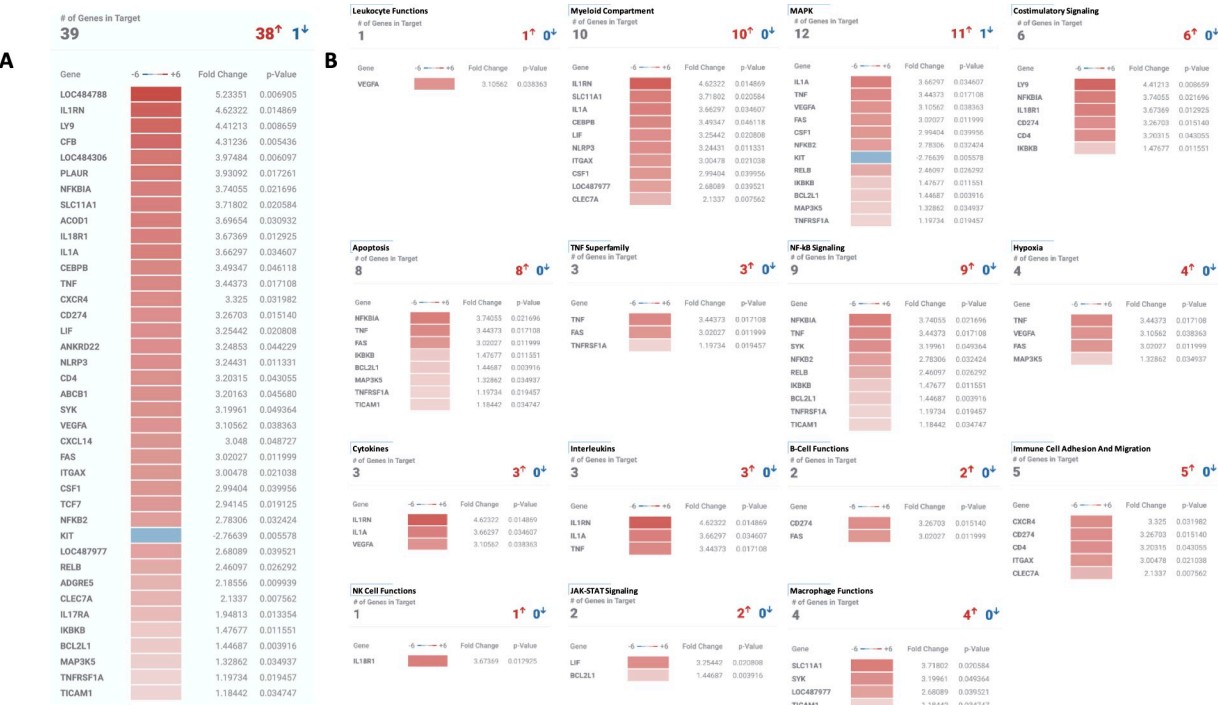

**Fig 6. Differential canine immuno-oncology gene expression at day 13 post initiation of therapy.** (A) Overall, 38 genes had significantly increased expression and 1 gene had significantly decreased expression compared to pretreatment samples. (B) Clustering of differential gene expression changes by biologically relevant immuno-oncology pathways.

well as time from blood collection to serum separation. Grade 1 and 2 hepatic enzyme (ALKP and ALT) elevations occurred in all companion dog treatment groups and likely has multifactorial attributions including repeated anesthetic events (all EBRT and PET/CT imaging was performed under general anesthesia), primary hepato-biliary excretion of [86/90]Y-NM600 [36], and IT-IC treatment (elevated hepatic enzyme levels have been documented with hu14.18-IL2 therapy in humans) [33]. As hepatic enzyme elevations were documented in all treatment groups, while of low grade, it would be prudent to monitor hepatic function in future studies.

Three of 15 dogs (20%) receiving IT-IC developed transient low-grade eosinophilia. Eosinophilia from IL2 therapy has been proposed as a mechanism of capillary leak syndrome [54] in humans which was not noted in this study population; however, a similar mechanism of IL2 induced eosinophilia appears to exist in dogs.

Non-radioactive NM600 at approximately 10-fold the maximum mass dosage subsequently used in companion dogs receiving [90]Y-NM600 was also found to be well tolerated in adult male and female purpose-bred beagles. For example, the companion dog receiving the highest prescribed dose of [90]Y-NM600 (Dog 1; 17.55 mCi) was the equivalent of 0.1755 mg NM600. One dog (Dog 2) experienced an acute hypersensitivity reaction during and immediately following IV injection. This was attributed to known hypersensitivity of some dogs to the Tween 20 excipient used [55].

Transient pain during IT-IC injection did require pharmacological restraint for administration in most patients. Delayed self-mutilation of the tumor site occurred in one dog following IT-IC alone and one dog following IT-IC/EBRT combination. It was not possible to determine if this was a reflection of irritation from either IT-IC and RT induced inflammation and sensory changes, or progression of disease.

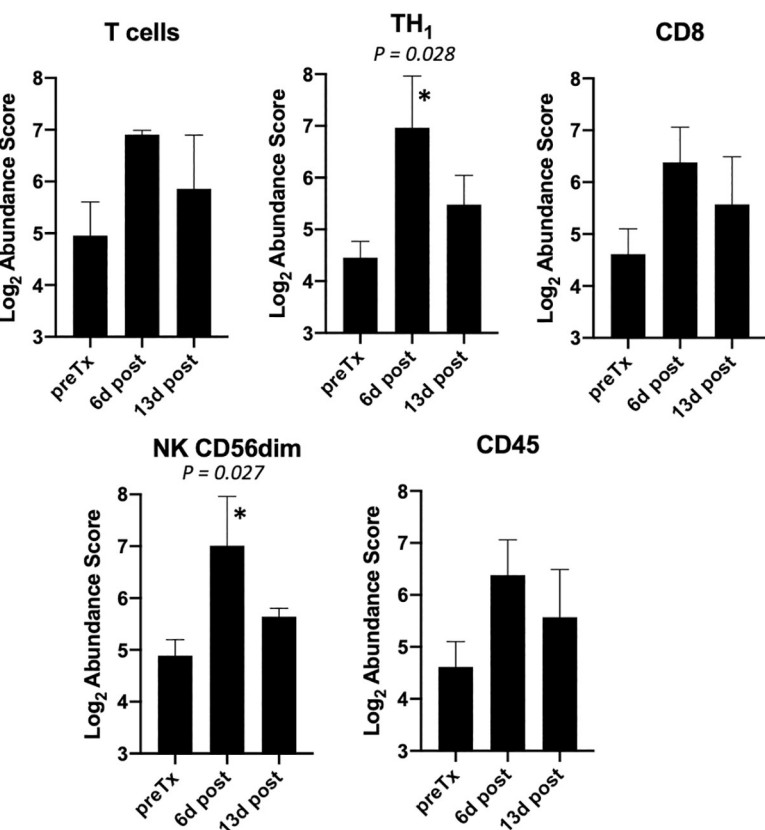

**Fig 7. Primary tumor cell type profiling before and after therapy initiation.** The cell abundance scores (Mean +/- SEM) for all dogs (n = 3) primary tumors in the experiment are displayed and grouped by cell type. Cell Type Scores are based on the NanoString Cell Type Profiling Module. *P values* are for non-parametric repeated measures ANOVA (Friedman test) and '*' identifies those time points deemed significantly different ($P < 0.05$) from pretreatment baseline by Dunn's multiple comparison analysis.

The ability of our theranostic pairing of [86]Y-NM600 and [90]Y-NM600 to document differential tumor uptake, allow tumor staging, and estimate subject-specific dosimetry leading to a safe prescribed dose of TRT in tumor-bearing companion dogs translated well from our earlier murine work. All but one dog demonstrated sufficient differential metastatic tumor:bone marrow uptake to deliver a dose of TRT theoretically capable of eliciting an immunomodulatory effect in the tumor and TME which could inhibit CIT without systemic immune suppression as demonstrated in our murine models [27, 28, 31]. Interestingly, in all dogs individually, the calculated absorbed dose of [90]Y-NM600 using serial [86]Y-NM600 PET/CT scans and RAPID Monte Carlo-based analysis revealed that the primary (index) tumor consistently had the lowest predicted absorbed dose of any of the systemic tumor sites. This may reflect that the primary tumor, in all but one case, represented an external site (oral, cutaneous) prone to necrosis and secondary inflammation or something inherently different about metastatic tumor cell NM600 incorporation. Regardless, since the primary/index tumor also receives EBRT, it is unnecessary to utilize this site for determination of the 2:1 differential uptake of metastatic tumor to bone marrow. Therefore, we only factor in the non-primary/index sites of lowest TRT tissue absorbed dose when determining tumor:bone marrow differential uptake. This should allow administration of a higher [90]Y-NM600 dose to metastatic tumors while maintaining safe absorbed dose to bone marrow.

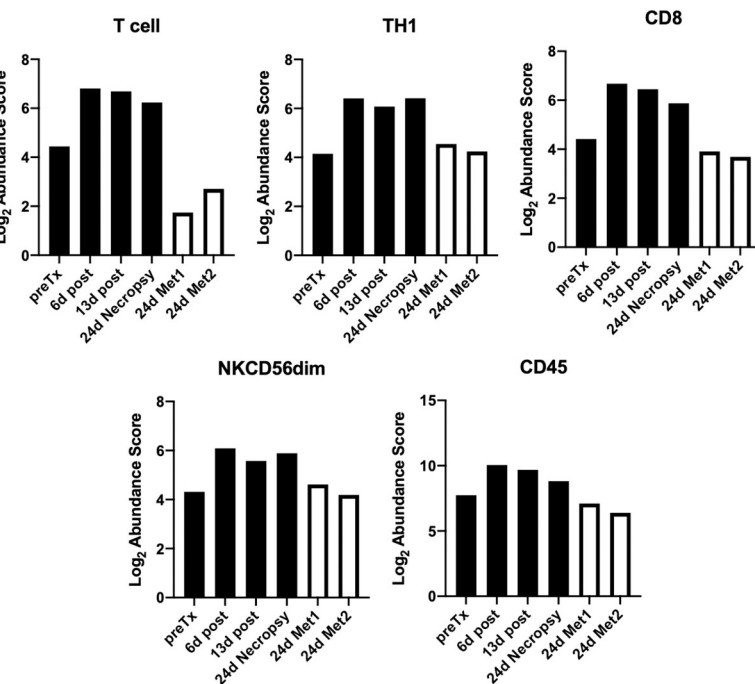

**Fig 8. Primary and metastatic tumor cell type profiling before and after therapy initiation in dog 4.** The cell abundance scores for the index tumor (solid bar) and two metastatic tumors (empty bar) are displayed and grouped by cell type. Cell Type Scores are based on the NanoString Cell Type Profiling Module.

All biospecimens have been banked for future analysis upon completion of higher TRT dosing cohorts currently underway. However, for an exploratory assessment of early immuno-modulatory effects, biospecimens from the subset of three dogs with melanoma that received the entire trimodal immune-radiotherapy protocol were preliminarily interrogated. While the number of dogs evaluated for immunologic changes resulting from therapy is too small and underpowered to make generalizations, the data does suggest immunomodulatory changes were manifested in primary tumor gene expression, TIL gene expression signatures and TIL numbers. Regarding gene expression changes, several pathways annotated to immune-oncology in the nCounter Canine IO Panel were determined to be upregulated 6 and 13 days

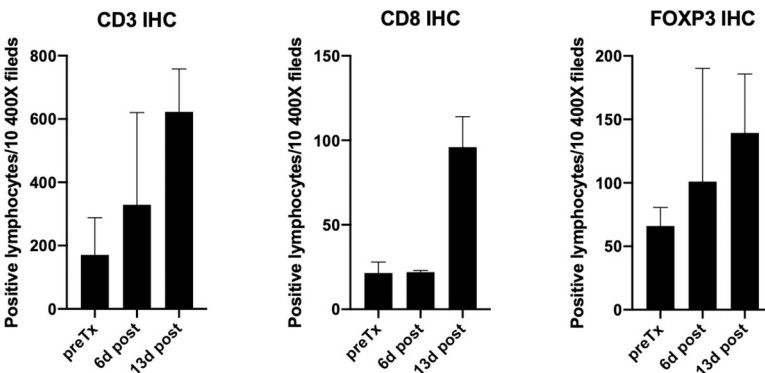

**Fig 9. Immunohistochemical assessment of tumor infiltrating lymphocyte populations in primary tumors before and during therapy.** Values represent the sum of the total lymphocyte distribution that occupied the tumor tissue in ten 400X fields (mean +/- SEM).

**Table 3. Composition of circulating T cells.**

| Lymphocyte Subset | PreTreatment (Mean +/-SEM) | Day 6 Post Treatment (Mean +/- SEM) | Day 29 (+/-3d) Post Treatment (Mean +/- SEM) |
|---|---|---|---|
| *CD3$^+$ | 42.52 +/- 2.23% | 36.86 +/- 14.93% | 20.89 +/- 12.28% |
| #CD3$^+$CD4$^+$ | 52.24 +/- 9.84% | 38.82 +/- 13.58% | 47.38 +/- 4.69% |
| #CD3$^+$CD8$^+$ | 29.35 +/- 11.46% | 46.72 +/- 14.40% | 30.89 +/- 4.73% |
| #Treg (CD3$^+$CD4$^+$CD25$^+$FoxP3$^+$) | 8.32 +/- 1.23% | 7.18 +/- 1.16% | 12.92 +/- 2.12% |
| *NK cells (CD3$^-$CD5$^{dim}$) | 3.35 +/- 0.69% | 3.63 +/- 1.03% | 2.01 +/- 0.64% |

The expression of T and NK cell subsets in the peripheral blood of 3 dogs with advanced stage melanoma at serial time points during multimodality immuno-radiotherapy.

* as a percentage of PBMCs.

#as a percentage of T lymphocytes.

following treatment initiation (Figs 5 and 6). These include pathways involved in T, B and NK cell function, as well as several costimulatory and signal transduction pathways. For example, a nearly 3-fold increase in TCF7 gene expression at day 13 post therapy was seen. In humans, TCF7 encodes the transcription factor T cell factor-1, and is a direct target of the Wnt and beta-catenin signaling axis. TCF7 has been observed in long lived Th17 cells [56] as a marker of stem cell memory [57]. Increased gene expression of TCF7 has been associated with stable beta-catenin as well as promotion of memory and survival programs while limiting terminal

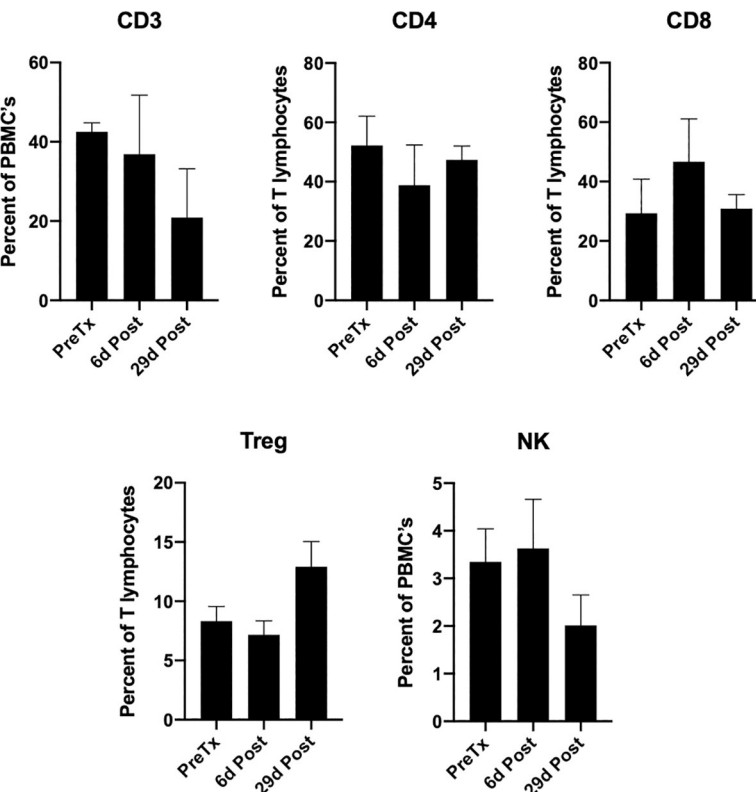

**Fig 10. Composition of circulating T cells.** The relative frequency of T and NK cell subsets (mean percent +/- SEM) in the peripheral blood of 3 dogs with advanced stage melanoma at serial time points during multimodality immuno-radiotherapy. Percentage of expression is based on cells within a lymphocyte gate described by a Forward versus Side Scatter flow cytometry plot.

effector differentiation. TCF7 expression also enriches for CD56[bright] NK cells [58] and is associated with memory-like NK cells during HIV infection [59]. In this model, induction of TCF7 gene expression at day 13 after *in situ* vaccination suggests persistent antitumor responses can be generated by this treatment regimen and may be associated with promotion of memory T/NK cell subsets. Not all the expression changes observed may be clinically beneficial. For example, a 3-fold increase in CTLA4 expression 6 days after initiation of therapy would intuitively dampen a cytotoxic immune response and may reflect Treg infiltration, which we observe following radiation therapy and may require anti-CTLA-4 mAb to overcome [28]. In our mouse models, checkpoint inhibition is required to see the full benefit of *in situ* vaccine and TRT therapy [28]. We have planned to incorporate checkpoint inhibition into future canine protocols as caninized checkpoint inhibitors become available including the newly conditionally-approved caninized anti-PD-1 monoclonal antibody, gilvetmab.

The relative abundance of expression signatures for several immune cell types (T, CD8, CD45, TH1, and NK CD56[dim)] increased in the primary tumors during therapy, with that of TH1 and NK CD56[dim] attaining statistical significance despite the small numbers (Fig 7). In human melanoma patients, CD56[dim] expressing NK cells have enhanced cytolytic activity and increased gene-expression signatures for this NK subset were associated with better clinical outcomes following immunotherapy [60]. Interestingly, in the one patient where TRT treated metastatic lesions were sampled along with the EBRT/IT-IC treated index tumor, while the relative abundance values for the cell types remained high in the index tumor, values in metastatic lesions were similar to the baseline values in the untreated index tumor (Fig 8), implying that metastatic sites did not experience the degree of immunomodulation observed in the index tumor. Similar to gene-signature cell type profiling, relative increases in CD3, CD8 and FoxP3 TILs were observed by IHC in primary tumors, although this was maximal at 13 days by IHC assessment. The timing of increased relative abundance of TIL cell profile expression signatures and IHC TIL numbers (i.e., peak at day 6–13) mimics observations we have reported in murine melanoma models for tumor immune susceptibility gene expression changes with EBRT [61] and with TRT [62] and cellular infiltrate changes with EBRT + IT-IC [27, 28].

This suggests relatively conserved time-scale for immunomodulatory effects of RT in mice and dogs. Flow cytometric analysis of circulating PBMC revealed numerical trends towards a decreased relative frequency of circulating T lymphocytes and NK cells and an increased relative frequency of Treg at day 29, however, this did not achieve statistical significance. A numerical increase in the relative frequency of CD8 lymphocytes occurred at day 6.

From these data, it is not possible, to determine which arm(s) of the trimodal immuno-radiotherapy protocol may be responsible for the immunomodulatory changes observed; attribution of therapy effects will await a more complete interrogation of the individual treatment groups (i.e., IT-IC alone, EBRT/ IT-IC and EBRT/TRT/IT-IC groups) as well as addition of currently accruing dogs into higher TRT dosing groups. Further, while we have documented preliminary evidence of immunomodulatory changes resulting from the trimodal immuno-radiotherapy protocol, any clinical or antitumor benefit from these changes remains theoretical at present. Interpretation will ultimately require greater case numbers with clinical follow-up.

There was no prerequisite for GD2 expression in tumors for entry into these proof-of-concept trials, however, *post hoc* analysis will be performed on all samples in all treatment groups at the completion of study. In the 3 dogs receiving the complete trimodal therapy, GD2 expression was documented in only one index tumor. This is consistent with expression levels in human patients with advanced melanoma accrued to a recent hu14.18-IL2 trial in which 6/12 patients were positive on *post-hoc* GD2 expression analysis [23]. No significant differences in

recurrence-free or overall survival were noted in that pilot trial in humans, however analyses of RNAseq gene signatures of immune activation in tumor specimens obtained ~ 2 weeks after receiving hu14.18-IL2 were significantly associated with event-free and overall survival in that study [63]. At present the number of canine patients in our data set remains too limited to assess any differences between immunologic change based on GD2 expression or to be able to assess for associations of these immune gene signatures with outcome. These may be more informative once accrual is complete to this trial. While post-hoc assessment of GD2 expression is acceptable for this proof-of-concept feasibility trial in dogs, future trials designed to assess antitumor efficacy should require GD2 expression for eligibility. The authors recognize a further limitation for the use of hu14.18-IL2 in canine trials is the likelihood of neutralizing antibodies against the xenogeneic human chimeric therapeutic mAb. However, the intratumoral delivery of hu14.18-IL2, as done here, is less susceptible to *in vivo* neutralizing effects of a mouse anti-hu14.18-IL2 antibody than is seen with intravenous administration [64]. If the canine model were to be used to investigate repeated hu14.18-IL2 treatments, this limitation could become important and development of caninized immunocytokines would likely be necessary.

Additional biospecimen analysis is planned once all cohorts are completed and will include quantifying TRT dose-dependent effects on tumor-infiltrating immune cells, expression of immune susceptibility markers, expression of markers of T cell exhaustion (PD-1, CTLA-4 and Tim3), effector function (IFN-γ and TNF) in the TME, and quantification of T cell receptor (TCR) diversity. Prior studies have shown a role for RT in diversifying the antitumor TCR repertoire when delivered in conjunction with immunotherapy [65] and TCR diversity may predict response [66, 67].

While antitumor efficacy and clinical outcomes were not primary endpoints in this proof-of-concept trial, one dog experienced a strong PR of the primary tumor with initial progression and then stabilization of the metastatic pulmonary nodule. While it is certainly possible the primary site response was due to EBRT alone, the continued stabilization of metastatic disease after a short progressive period is intriguing and would be categorized as immune unconfirmed PD (iUPD) based on iRECIST guidelines [68]. The other 3 dogs, all with advanced stage IV disease, experienced early progression of metastatic lesions. Given the median survival time for advanced cancer in dogs is reported to be 2–3 months even with standard of care therapy [69, 70], the mortality rate in this study population is not unexpected. Progression in two of these dogs was a result of brain metastasis. Neither lesion was observed on [86]Y-NM600 PET/CT scans a month prior to clinical documentation, implying either the lesions were indeed rapidly advancing new lesions, or that they were present but in the context of an intact blood brain barrier (BBB) at the time of PET/CT. Cancer-targeting APCh analogues have been shown to demonstrate poor permeability across intact BBB [71] and may not be suitable for TRT-based staging or treating intracranial disease in this context.

Future directions for our work include continued dose escalation for TRT using the combination *in situ* vaccination and TRT outlined in this study and to characterize dose-dependent immunomodulatory effects and antitumor efficacy of the *in situ* vaccination combined with systemic TRT in larger cohorts. Indeed, the 3 dogs with melanoma receiving the complete EBRT/IT-IC/TRT protocol in the study presented here represent the first cohort in an ongoing TRT dose escalation trial (U01). We also plan to incorporate checkpoint inhibition (once caninized products are available), into our protocol as we have shown in murine models that their addition provides further enhancement of antitumor efficacy [28]. Furthermore, while the *in situ* vaccine/TRT combination results in enhanced TCR diversity, checkpoint inhibition is required for T cell expansion at the site of tumor [72]. Further, we plan to investigate the use of alternative radionuclides (e.g., actinium-225, lutetium-177) chelated with NM600 in similar

tumor-bearing companion dog models, as they present radioactive decay properties that may be more advantageous for immunomodulation than those of $^{90}$Ytrrium (i.e., longer half-life, shorter particle range, and high linear energy transfer).

## Conclusions

The combination of EBRT/IT-IC/TRT immuno-radiotherapy known to have activity against syngeneic melanoma in murine models is feasible and well tolerated in this small cohort of companion dogs with advanced stage, spontaneously arising melanoma or osteosarcoma. Early evidence for immunomodulation was observed, however interpretation of these changes awaits additional study. Further expansion and characterization of the immunologic effects of this approach in a relevant immunocompetent large animal surrogate with clinically relevant heterogenous tumor/TME's similar to humans should better recapitulate the human condition, cross-validate our mouse data and serve as an important translational bridge to planned human clinical trials.

## Supporting information

**S1 Fig. Flow cytometric analysis of GD2 expression in canine and human tumor cells.** Thick black line indicates cells positively stained with anti-GD2-PE (clone 14.G2a, a murine IgG2a anti-GD2 mAb) and grey shaded areas indicate FMO control. Values indicate percent GD2$^{+}$ cells. 17CM98, CML-6M and CML-1 are canine melanoma cell lines. M21 is a human GD2$^{+}$ melanoma cell line used as a positive control. Jurkat is a human GD2$^{-}$ lymphoma cell line used as a negative control.
(TIF)

**S2 Fig. GD2 Expression in canine primary tumors.** Frozen sections of canine tissues labeled with anti-GD2 mAb (clone 14.G2a, a murine IgG2a anti-GD2 mAb) conjugated to PE (red) and counter-stained with DAPI (blue). A, Canine spleen negative control; B, canine oral malignant melanoma; C, canine soft tissue sarcoma.
(TIF)

**S3 Fig. Representative example of gating strategy for flow cytometric analysis of PBMCs.**
(TIF)

**S4 Fig. Expression levels of spike-in samples.** A, positive and negative controls. B, House-keeping genes. The square root of the expression level is used to show the lower expression values. Note that specimen legend identifiers are coded and can be found in S3 Table.
(TIF)

**S5 Fig. Sample correlation heatmap.** A sample correlation heatmap providing a graphical representation of data, in which the individual values contained in the matrix are represented as colors. In this case, the data matrix contains correlation values between samples, with the darkest blue representing the strongest correlation. The dendrogram annotation on the top axis provides information regarding the clustering of samples. Samples that are closely related (i.e., those in the same replicate group) are strongly correlated together in the plot and are the closest branches of the dendogram. Note that specimen legend identifiers are coded and can be found in S3 Table.
(TIF)

**S6 Fig. Violin plot displaying the distribution of the log of the un-normalized gene counts for each sample in the experiment.** Note that specimen legend identifiers are coded and can

be found in S3 Table.
(TIF)

**S7 Fig. Variance of Mean plot.** This plot maps the variance mean of the log2 expression of all targets and probes in the panel. Housekeeping probes are colored to indicate which ones were and were not used in normalization.
(TIF)

**S1 Table. Immunohistochemical reagents and parameters.**
(DOCX)

**S2 Table. Flow cytometric reagents and parameters.** APC, allophycocyanin; FITC, fluorescein; PE, phycoerythrin; PE-Cy7, PE-cyanine 7; PerCP-eFluor710, peridinin chlorophyll protein-eFluor$^{TM}$710; SB60, Super Bright 600.
(DOCX)

**S3 Table. RNA quality of samples.**
(DOCX)

## Acknowledgments

We gratefully acknowledge Archeus Technologies (Madison, WI) for provision of NM600 and Apeiron Biologics (Vienna, Austria) for provision of the hu14.18-IL2 fusion immunocytokine used in these studies. The authors thank the University of Wisconsin Translational Research Initiatives in Pathology laboratory (TRIP), supported by the UW Department of Pathology and Laboratory Medicine, for IHC and Nanostring analysis. We also acknowledge Dr. David Gasper for providing the IHC reads for TIL analysis.

## Author Contributions

**Conceptualization:** Ian R. Marsh, Joseph Grudzinski, Ilene D. Kurzman, Cindy L. Zuleger, Mark R. Albertini, Bryan Bednarz, Paul M. Sondel, Jamey P. Weichert, Zachary S. Morris, Reinier Hernandez, David M. Vail.

**Data curation:** Kara Magee, Ian R. Marsh, Michelle M. Turek, Eduardo Aluicio-Sarduy, Jonathan W. Engle, Christine Jaskowiak, Karla Esbona, Zachary S. Morris.

**Formal analysis:** Ian R. Marsh, Michelle M. Turek, Joseph Grudzinski, Eduardo Aluicio-Sarduy, Jonathan W. Engle, Ilene D. Kurzman, Cindy L. Zuleger, Christine Jaskowiak, Mark R. Albertini, Karla Esbona, Bryan Bednarz, Paul M. Sondel, Zachary S. Morris, Reinier Hernandez, David M. Vail.

**Funding acquisition:** Mark R. Albertini, Jamey P. Weichert, Zachary S. Morris, David M. Vail.

**Investigation:** Kara Magee, Michelle M. Turek, Joseph Grudzinski, Cindy L. Zuleger, Elizabeth A. Oseid, Mark R. Albertini, Bryan Bednarz, Zachary S. Morris, Reinier Hernandez, David M. Vail.

**Methodology:** Michelle M. Turek, Joseph Grudzinski, Ilene D. Kurzman, Cindy L. Zuleger, Elizabeth A. Oseid, Christine Jaskowiak, Mark R. Albertini, Karla Esbona, Bryan Bednarz, Paul M. Sondel, Jamey P. Weichert, Zachary S. Morris, Reinier Hernandez, David M. Vail.

**Project administration:** Ilene D. Kurzman, Elizabeth A. Oseid, Mark R. Albertini, Zachary S. Morris, David M. Vail.

**Resources:** Zachary S. Morris, David M. Vail.

**Software:** Ian R. Marsh, Bryan Bednarz.

**Supervision:** Ilene D. Kurzman, Christine Jaskowiak, Mark R. Albertini, Zachary S. Morris, David M. Vail.

**Validation:** Ian R. Marsh, Michelle M. Turek, Joseph Grudzinski, Eduardo Aluicio-Sarduy, Jonathan W. Engle, Ilene D. Kurzman, Cindy L. Zuleger, Karla Esbona, Bryan Bednarz, Paul M. Sondel, Jamey P. Weichert, Zachary S. Morris, Reinier Hernandez, David M. Vail.

**Visualization:** Jonathan W. Engle, Karla Esbona, Jamey P. Weichert, Zachary S. Morris, Reinier Hernandez, David M. Vail.

**Writing – original draft:** David M. Vail.

**Writing – review & editing:** Kara Magee, Ian R. Marsh, Michelle M. Turek, Joseph Grudzinski, Eduardo Aluicio-Sarduy, Jonathan W. Engle, Ilene D. Kurzman, Cindy L. Zuleger, Elizabeth A. Oseid, Christine Jaskowiak, Mark R. Albertini, Karla Esbona, Bryan Bednarz, Paul M. Sondel, Jamey P. Weichert, Zachary S. Morris, Reinier Hernandez.

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
