## [Decision Letter · Decision Letter 0]

16 Feb 2021

PONE-D-21-01036

Safety and feasibility of an in situ vaccination and immunomodulatory targeted radionuclide combination immuno-radiotherapy approach in a comparative (companion dog) setting

PLOS ONE

Dear Dr. Vail,

Thank you for submitting your manuscript to PLOS ONE. After careful consideration, we feel that it has merit but does not fully meet PLOS ONE’s publication criteria as it currently stands. Therefore, we invite you to submit a revised version of the manuscript that addresses the points raised during the review process.

Please address all Reviewer comments.

We look forward to receiving your revised manuscript.

Kind regards,

Douglas H. Thamm, V.M.D.

Academic Editor

PLOS ONE

Journal Requirements:

2. As part of your revision, please complete and submit a copy of the Full ARRIVE 2.0 Guidelines checklist, a document that aims to improve experimental reporting and reproducibility of animal studies for purposes of post-publication data analysis and reproducibility: https://arriveguidelines.org/sites/arrive/files/Author%20Checklist%20-%20Full.pdf (PDF). Please include your completed checklist as a Supporting Information file. Note that if your paper is accepted for publication, this checklist will be published as part of your article.

"I have read the journal's policy and the authors of this manuscript have the following competing interests:

ZM – Scientific Advisory Board Member and equity options, Archeus Technologies, Scientific Advisory Board Member and equity options, Seneca Therapeutics, Research Material support (reagents) from Bristol-Myers Squibb, AstraZeneca, Nektar Therapeutics, Apeiron Biologics, XRAd Therapeutics.  BB and JG has ownership interest in Voximetry Inc (Middleton, WI).  MRA - Research collaborations through the University of Wisconsin with Bristol Myers Squibb (Redwood City, CA and Princeton, NJ) and with Apeiron Biologics (Vienna, Austria). JW is a co-founder and CSO of Archeus Technologies, Inc (Madison, WI)."

Reviewers' comments:

Reviewer's Responses to Questions

**Comments to the Author**

1. Is the manuscript technically sound, and do the data support the conclusions?

Reviewer #1: Yes

Reviewer #2: Yes

2. Has the statistical analysis been performed appropriately and rigorously? 

Reviewer #1: I Don't Know

Reviewer #2: N/A

3. Have the authors made all data underlying the findings in their manuscript fully available?

Reviewer #1: Yes

Reviewer #2: Yes

4. Is the manuscript presented in an intelligible fashion and written in standard English?

Reviewer #1: Yes

Reviewer #2: Yes

5. Review Comments to the Author

Reviewer #1: This is an interesting study assessing a combinatorial approach of immuno-radiotherapy in metastatic disease using companion canine patients. While the primary end-points centered on possible toxicities (of which there were minimal) and feasibility, the study suffers in the lack of any immunological or tumor analysis data essential for being able to draw any conclusions on whether the regimen was having any immune or tumor effects (and importantly, if the regimen needs to be developed more as no DLT was observed on the single treatments especially given the extremely small sample size) and heterogeneity of the disease/breeds.

There is a need for both local and systemic immune parameters (especially T and NK cells, serum cytokines etc) to be shown including during treatment and post-mortem when available. Extensive or in-depth assessments are not needed but given the preclinical and clinical use of of the immunokine and the species divergence, it is essential to include immune parameters in the study. Otherwise there is little data and insights on determining if key parameters (ie immune) are even being impacted by the regimens given.

Other points:

1) the use of hu14.18-IL2 is at first confusing versus simply using IL2 as it is not expected that the anti-human GD2 mAb used will bind to canine GD2 at all. There needs to be much more background on this limitation and immunokine (how does the IL2 IU activity compare? does any cross-reactivity exist? Neutralizing responses (as totally human and therefore xenogeneic) by the patient? The discussion should include limitations on the canine model in these cases and draw also from the clinical data with this agent.

2) there needs to be more information on the disease status and health of the client-owned dogs as well as information on prior treatments, if any as all impact toxicities.

Reviewer #2: This is a very interesting pilot project to demonstrate safety and feasibility of a trimodal approach to therapy for metastatic cancer. The progression of the safety assessment is logical and generally well described. The approach is very novel and the general methods described robust for the intended project. Some description of methods is absent and would be better included.

Comments:

Abstract: please include the number of dogs in the abstract.

Introduction:

You properly use the noun "dogs" in the abstract but include "canines" in the rest of the manuscript. Most veterinary journals prefer "dogs" as the noun and "canine" as the adjective.

Materials and Methods:

Please include the IACUC protocol number.

The radiolabeling protocol is not described. Please briefly describe it and highlight any differences in protocol (if present) from the referenced article.

What is the source and production methods of the Y-90 used?

Results:

The term "strong partial" is difficult to quantitate. The result in the described dog was a partial response of 67% reduction.

In the normal dog safety study, please include attribution with the hypo- and hyperglycemia as you do for the hematuria.

The tumor to bone marrow ratio calculation is difficult to understand in both the methods and the result. Looking at the values in Figure 3, it appears that only one dog actually had a primary tumor:bone marrow ratio > 2. This is explained somewhat further in the Discussion, but warrants clarification in the methods and results.

Figures 1 and 2 were difficult to read in the version this reviewer received.

If the immune infiltrates or peripheral blood immune populations have been analyzed, the results should be included in this manuscript to round out preliminary evidence of effect along with feasibility.

Discusion:

The discussion is reasonable.

6. PLOS authors have the option to publish the peer review history of their article (what does this mean?). If published, this will include your full peer review and any attached files.

Reviewer #1: No

Reviewer #2: No

---

## [Author Response · Author response to Decision Letter 0]

25 Jun 2021

June 21th, 2021

Editors, Plos One

RE: Response to Reviewers - manuscript PONE-D-21-01036.

Dear Editor:

We greatly appreciate the thoughtful comments and suggestions of the two reviewers and feel the changes and additions we have made serve only to make the manuscript better. In particular, while we had initially made a decision to exclude any treatment-related immunological assessments in this feasibility presentation, we have now added exploratory assessments of immunomodulatory changes in tumor gene expression levels (nCounter NanoString), tumor infiltrating lymphocyte subsets (nCounter cell type profiling, and IHC analysis) and peripheral circulating cell subset analysis (flow cytometric characterizations). These explorations, based on the first multi-modal immuno-radiotherapy cohort (n = 3) in our ongoing studies, provide preliminary evidence for immunomodulatory effects of our protocol which, when expanded in our ongoing studies should provide valuable data as to the mechanisms involved, allow for further modifications of protocol and characterize the utility of the surrogate system. We required an extension of the revision deadline to complete this exploration due to ongoing reagent availability and laboratory personnel density limits resulting from COVID restrictions at our institution.

Authors Response to Reviewers Queries:

Reviewer #1:

General Comment: “…the study suffers in the lack of any immunological or tumor analysis data essential for being able to draw any conclusions on whether the regimen was having any immune or tumor effects (and importantly, if the regimen needs to be developed more as no DLT was observed on the single treatments especially given the extremely small sample size) and heterogeneity of the disease/breeds. There is a need for both local and systemic immune parameters (especially T and NK cells, serum cytokines etc) to be shown including during treatment and post-mortem when available. Extensive or in-depth assessments are not needed but given the preclinical and clinical use of the immunocytokine and the species divergence, it is essential to include immune parameters in the study.”

We had initially presented only safety and feasibility data for this proof-of-concept first cohort while batching biospecimens (tumor/normal/PBMC, serum/plasma) samples for immune-analysis once ongoing expanded cohorts with increasing TRT doses were completed; however, we certainly agree that this was a deficiency in the initial manuscript. As COVID restrictions have relaxed somewhat in our laboratories, we now present, as requested, exploratory characterizations of immunomodulatory effects of our combined immune-radiotherapy approach in our first melanoma cohort (n=3). These include use of the nCounter Canine Immuno-Oncology (IO) gene expression panel that identified gene expression changes occurring one and 2 weeks after therapy as well as changes in tumor infiltrating lymphocyte populations by both nCounter cell type profiling capabilities and standard IHC analysis. Further we characterized peripheral circulating immune cell subset by flow cytometric characterizations at similar time points. These data are now presented in new figures and described in detail in methods and results under the subheadings ‘Exploratory assessment of immunomodulatory effects’ in addition to substantial text added to the discussion section pertaining to them. 

While numbers are small these explorations reveal immunologic changes occurring over time resulting from our protocol. Confident and in-depth interpretation of these changes will ultimately require further interrogation of expanded cohorts and follow-up clinical data which are the subject of ongoing endeavors. Additionally, the effect to which each of the trimodal treatments (i.e., IT-IC, EBRT, and TRT) contributed to the observed immunomodulatory effects cannot be discerned from this data set. Planned interrogations on biospecimens collected from dogs receiving single-agent IT-IC, EBRT and IT-IC, and comparison to the trimodal (EBRT/TRT/IT-IC) groups (current and ongoing) will be necessary for such an assessment.

Other points:

1) the use of hu14.18-IL2 is at first confusing versus simply using IL2 as it is not expected that the anti-human GD2 mAb used will bind to canine GD2 at all. There needs to be much more background on this limitation and immunokine (how does the IL2 IU activity compare? does any cross-reactivity exist? Neutralizing responses (as totally human and therefore xenogeneic) by the patient? The discussion should include limitations on the canine model in these cases and draw also from the clinical data with this agent.

Our group has previously worked with the hu14.18-IL2 immunocytokine in dogs and canine tissues. As presented in the original manuscript, reference 24 and supplemental figures 1 & 2 describes our groups work documenting GD2 expression in canine melanoma cell cultures and canine melanoma and sarcoma primary tumors by flow cytometry and immunofluorescence microscopy, respectively. Further, our groups work (ref 24) has documented that both the mouse 14.G2a monoclonal (used in our diagnostic flow and immunofluorescence microscopy assays) and the mouse-human chimeric 14.18 anti-GD2 mAb (used in our clinical fusion protein) enhanced lysis of canine melanoma cells by canine peripheral blood lymphocytes indicating their ability to direct ADCC by canine effector cells. Additionally, our group (ref 25) and several others have has also shown that human IL2 has significant immunologic effects in dogs and that the addition of human IL2 augments the ADCC potentiation of the 14.18 anti-GD2 mAb against canine melanoma cells (ref 24) as is the case in human melanoma. As GD2 is a disialoganglioside, it is not species specific, and the 14.18 anti-GD2 mAb can recognize mouse, human and canine melanoma. We have clarified this in the paragraph beginning on line 107 of the revision (track-changes version). While there was no a priori requirement for GD2 expression in the proof-of-concept trials outlined in the manuscript, we plan on assessing this in batch at study completion. However, in the revised manuscript we did perform immunofluorescent microscopy on the index tumors in the 3 dogs with melanoma receiving the complete trimodal immuno-radiotherapy protocol (EBRT/TRT/IT-IC) and found 1 of 3 to express GD2. This is in line with human melanoma patients (ref 23) where 6 of 12 patents were GD2 positive. We have added this to the manuscript and addressed in the discussion. Regarding neutralizing responses (since it is a xenogeneic mAb), as requested, we have added new text to the discussion section (see the text at the end of the paragraph starting on line 773) as a limitation to recurring or multiple hu14.18-IL2 treatments and that caninized platforms would likely be necessary for repeated treatments in future studies. 

2) there needs to be more information on the disease status and health of the client-owned dogs as well as information on prior treatments, if any as all impact toxicities.

All dogs had measurable disease of at least 2 cm longest diameter (line 180). The tumor clinical stage for dogs in the EBRT/IT-IC combination protocol are listed on line 473 in results. Further, the clinical stage for dogs in the EBRT/TRT/IT-IC combination protocol are listed in the manuscript in table 2 in results. Regarding general health, a statement that all dogs had to have a pretreatment constitutional clinical sign status of 0 or 1 (normal or asymptomatic/mild symptoms not requiring intervention) according to VCOG CTCAE v1.1 at entry was added to line 176. Regarding prior treatments, exclusion requirements (now listed in Line 178) dis-allowed prior radiation therapy or immunotherapy and required a minimum 2-week washout from previous chemotherapy. Prior surgery was allowable as long as post-surgical recurrence met the 2 cm minimum longest diameter cut-off. 

Reviewer #2:

1. Please include the number of dogs in the abstract.

Added as requested.

2. Prefer "dogs" as the noun and "canine" as the adjective.

 Corrected throughout.

3. Please include the IACUC protocol number.

 Added to line 166 in methods.

4. The radiolabeling protocol is not described. Please briefly describe it and highlight any differences in protocol (if present) from the referenced article.

 The protocol was added as requested on lines 262-269.

5. What is the source and production methods of the Y-90 used?

90Y was purchased from Perkin Elmer as 90YCl3 – a statement to this effect was added to methods (line 261)

6. The term "strong partial" is difficult to quantitate. The result in the described dog was a partial response of 67% reduction.

 The term “strong” was deleted as requested.

7. In the normal dog safety study, please include attribution with the hypo- and hyperglycemia as you do for the hematuria.

As in the discussion, the following sentence was added to attribute the hypoglycemia (line 494). “Low grade fluctuations in blood glucose levels in these dogs was likely attributable to a lack of standardization in feeding schedule and time from blood collection to serum separation.” 

8. The tumor to bone marrow ratio calculation is difficult to understand in both the methods and the result. Looking at the values in Figure 3, it appears that only one dog actually had a primary tumor:bone marrow ratio > 2. This is explained somewhat further in the Discussion, but warrants clarification in the methods and results.

We have clarified in the methods and results that the 2:1 tumor:bone marrow ratio requirement is for metastatic tumor:BM, not primary tumor:BM as the primary tumor receives 8Gy EBRT in the protocol and therefore is not considered relevant to the 2:1 ratio eligibility cut-off (line 298). Therefore, indeed 4 of 5 dogs met criteria and the one dog not meeting this criteria did not go on to receive TRT.

9. Figures 1 and 2 were difficult to read in the version this reviewer received.

We defer to the editors as fig 1 & 2 in our figure files are readable and we are not sure which versions were provided the reviewer. 

10. If the immune infiltrates or peripheral blood immune populations have been analyzed, the results should be included in this manuscript to round out preliminary evidence of effect along with feasibility.

Please see response to reviewer #1’s general query. We have added gene expression changes (nCounter Canine Immuno-oncology [IO] gene expression panel) occurring one and 2 weeks after therapy as well as changes in tumor infiltrating lymphocyte populations by both nCounter cell type profiling capabilities and standard IHC analysis. Further we characterized changes in peripheral circulating immune cell subset by flow cytometric characterizations at similar time points. These data were added in addition to further methodology and discussion points pertaining to them.

---

## [Decision Letter · Decision Letter 1]

26 Jul 2021

Safety and feasibility of an in situ vaccination and immunomodulatory targeted radionuclide combination immuno-radiotherapy approach in a comparative (companion dog) setting

PONE-D-21-01036R1

Dear Dr. Vail,

We’re pleased to inform you that your manuscript has been judged scientifically suitable for publication and will be formally accepted for publication once it meets all outstanding technical requirements.

Kind regards,

Douglas H. Thamm, V.M.D.

Academic Editor

PLOS ONE

Additional Editor Comments (optional):

Reviewers' comments:

Reviewer's Responses to Questions

**Comments to the Author**

1. If the authors have adequately addressed your comments raised in a previous round of review and you feel that this manuscript is now acceptable for publication, you may indicate that here to bypass the “Comments to the Author” section, enter your conflict of interest statement in the “Confidential to Editor” section, and submit your "Accept" recommendation.

Reviewer #1: All comments have been addressed

Reviewer #2: All comments have been addressed

2. Is the manuscript technically sound, and do the data support the conclusions?

Reviewer #1: Yes

Reviewer #2: Yes

3. Has the statistical analysis been performed appropriately and rigorously? 

Reviewer #1: Yes

Reviewer #2: Yes

4. Have the authors made all data underlying the findings in their manuscript fully available?

Reviewer #1: Yes

Reviewer #2: Yes

5. Is the manuscript presented in an intelligible fashion and written in standard English?

Reviewer #1: Yes

Reviewer #2: Yes

6. Review Comments to the Author

Reviewer #1: all major issues satisfactorily addressed. The authors have done a nice job in revising the manuscript.

Reviewer #2: (No Response)

7. PLOS authors have the option to publish the peer review history of their article (what does this mean?). If published, this will include your full peer review and any attached files.

Reviewer #1: No

Reviewer #2: **Yes: **Jeffrey Bryan, DVM, MS, PhD, DACVIM(Oncology)

---

## [Editor Report · Acceptance letter]

28 Jul 2021

PONE-D-21-01036R1 

Safety and feasibility of an *in situ* vaccination and immunomodulatory targeted radionuclide combination immuno-radiotherapy approach in a comparative (companion dog) setting 

Dear Dr. Vail:

I'm pleased to inform you that your manuscript has been deemed suitable for publication in PLOS ONE. Congratulations! Your manuscript is now with our production department. 

Kind regards, 

on behalf of

Dr. Douglas H. Thamm 

Academic Editor

PLOS ONE